# How much can we save? Impact of different emission scenarios on future snow cover in the Alps.

Christoph Marty[1], Sebastian Schlögl[1,2], Mathias Bavay[1], Michael Lehning[1,2]

[1]WSL Institute for Snow and Avalanche Research SLF, 7260 Davos, Switzerland
[2]Laboratory of Cryospheric Sciences CRYOS, EPFL, 1015 Lausanne, Switzerland

*Correspondence to*: Christoph Marty (marty@slf.ch)

**Abstract.** This study focuses on an assessment of the future snow depth for two larger Alpine catchments. Automatic weather station data from two diverse regions in the Swiss Alps have been used as input for the Alpine3D surface process

model to compute the snow cover at 200 m horizontal resolution for the reference period (1999-2012). Future temperature and precipitation change have been computed from 20 downscaled GCM-RCM chains for three different emission scenarios, including one intervention scenario (2°C target) and for three future time periods (2020-2049, 2045-2074, 2070-2099). By applying simple daily change values to measured time series of temperature and precipitationsmall-scale climate scenarios have been calculated for the  median estimate and extreme changes. The projections reveal a decrease in snow depth for all

elevations, time periods and emission scenarios. The non-interventions scenarios demonstrate a decrease of about 50 % even for the elevations above 3000 m. The most affected elevation zone for climate change is located below 1200 m, where the simulations show almost no snow towards the end of the century. Depending on the emission scenario and elevation zone the winter season starts half a month to one month later and ends one to three month earlier in this last scenario period. The resultant snow cover changes may roughly be equivalent to an elevation shift of 500-800 m or 700-1000 m for the two non-

intervention emissions scenario. At the end of the century the number of snow days may be more than halved at an elevation of around 1500 m and is predicted to only 0-2 snow days in the lowlands. The results for the intervention scenario reveal no differences for the first scenario period, but clearly demonstrate a stabilization thereafter, comprising much lower snow cover reductions towards the end of the century (ca. 30 % instead of 70 %).

## 1 Introduction

The inter-annual snow volume is highly variable in the Alps. This is mainly caused by the combined effect of temperature and highly variable precipitation amounts (Bartolini et al., 2009). Consecutive winters with little snow or short snow duration can endanger the livelihood of tens of thousands because up to 90 % of the economy in Alpine villages depends on winter tourism (Abegg et al., 2007), whereas huge amounts of snow can cause destructive avalanches like in winter 1999 (Wilhelm et al., 2001). The Alps are already affected by climate change, mainly by increasing temperatures (Serquet et al.,

2011). Several studies demonstrate the effect of these changes on the snow depth and snow duration(Scherrer et al.,

2004;Durand et al., 2009;Marty, 2008), on snow fall (Valt and Cianfarra, 2010;Serquet et al., 2011) or snow water equivalent (Marty et al., 2017).

Projections of the future winter climate reveal that temperatures will further increase whereas the precipitation signal in the Alps is less clear (Gobiet et al., 2014;Zubler et al., 2014), . On the point scale Schmucki et al. (2015a) have shown that the depth of future snow pack is clearly controlled by increasing temperatures and that the projected small increases in precipitation can only at highest elevations and only partly compensate the effect of the dominating warming signal. On the regional/catchment scale so far the impact of these changes on the snow cover has mostly been investigated by either using GCM / RCM output directly with the limited ability to model high elevation changes (Mankin and Diffenbaugh, 2015) given the coarse spatial resolution of such models or by using a limited set of high resolution RCMs and emission scenarios: For example, Steger et al. (2013) assessed the trend of future snow water equivalent (SWE) in the Alps from direct RCM output. Laghari et al. (2012) investigated the change of SWE and runoff in a catchment in the Austrian Alps by applying the conceptual hydrological model PREVAH to direct projections of a larger set of RCMs from the PRUDENCE project. Rousselot et al. (2012) modelled snowpack scenarios across the French Alps with the snow model CROCUS using the French RCM ALADIN. Marke et al. (2014) used the model AMUNDSEN with three RCMs and one emission scenario to simulate the future snow cover and ski season length for a small region in the central Austrian Alps.

The goal of this study is to investigate the impact of climate change on the Alpine snow cover with the surface process model Alpine3D (Lehning et al., 2006). In contrast to Bavay et al. (2009 & 2013), which used Alpine3D to analyse the changes in runoff in eastern Switzerland, we use a full set of RCM simulations, mean and extreme changes, three emissions scenarios and two diverse regions to investigate the difference in snow cover between two non-intervention scenarios and a climate stabilization scenario that supposes that global emissions are cut by about 50 % by 2050. This scenario likely limits global warming to 2°C since the preindustrial level. The two regions and the input data from meteorological stations are introduced in chapter 2.1 & 2.2. We use projections from 20 GCM-RCMs model combinations of the ENSEMBLES project (chapter 2.4) to perturb the current conditions determined by the data of the meteorological stations. These modified data were used to force the Alpine3D model (chapter 2.3) to simulate changes of the snow depth for different elevations and seasons.

## 2 Data and Methods

### 2.1 Investigated regions

Two different catchments (the Aare region in central Switzerland and the Grisons region in eastern Switzerland) were chosen to assess the future evolution of the Alpine snow pack (Figure 1). The Aare region comprises 3190 km$^2$ and its mean elevation is 1530 m asl. It is characterized by the Swiss Plateau in the north (mean altitude 500 m asl) and a small band of high mountains in the south. The precipitation amount in this region is mainly controlled by large scale weather patterns

coming from the northern Atlantic. The river Aare acts as the discharge of the catchment area and drains into the river Rhine and finally into the North Sea.

The Grisons region is three times larger (10373 km$^2$) and its mean elevation is 1903 m asl. It is characterized by a high alpine environment and a maximum elevation of 4049 m asl (Piz Bernina). The precipitation in its northern part is also

controlled by weather from the north-western sector, whereas the southern part is dominated by moisture from the Mediterranean Sea. The region drains mostly into the upper Rhine (North Sea) and the Inn (Black Sea). Some southern parts drain into the river Po and finally into the Adriatic Sea.

## 2.2 Input data

The meteorological data were provided by 48 automatic weather stations (AWS) in the Aare region and 34 AWS in the

Grisons region at hourly resolution. For both regions the large majority of the stations were located between 500 and 2500 m asl. The stations were selected based on the requirement that they provide hourly meteorological data and are located in or close to the simulation region. The following meteorological parameters were used for model input: Air temperature, relative humidity, wind velocity, precipitation, shortwave radiation and (if available) snow surface temperature and ground surface temperature. Incoming longwave radiation was parameterized and precipitation was corrected for wind-induced under-catch

as described in Schlögl et al. (2016). The years between 1999 and 2012 were selected as reference period in order to keep the data availability optimal. This 13 years period contains one of the most snow abundant (2009) and one of the most snow-scare winters (2007) in the last 30 years. The fact that not all stations provided the same number of parameters was not a problem since each parameter was individually interpolated to the model grid (see next chapter).

## 2.3 Alpine3D

The snow pack was simulated with the surface process model Alpine3D. Alpine3D is a spatially distributed (surface), three-dimensional (atmospheric) model for analysing and predicting the dynamics of snow-dominated surface processes in mountainous topography. It consists of a snow cover (SNOWPACK) optional modules like runoff, vegetation, soil and snow transport (Lehning et al., 2006). Essential input data for the successful simulation were the following different surface grids

and time series of meteorological parameters: A digital elevation model (DEM) with a horizontal resolution of 200 m was used in order to represent the topography of the two regions. The necessary land cover data were taken from CORINE (Bossard et al., 2000) with a horizontal resolution of 100 m and converted into the PREVAH classification (Viviroli et al., 2009). The PREVAH classification is less detailed than the CORINE classification, but sufficient for our simulations. These data were interpolated to the 200 m grid of the DEM by the nearest neighbour method. Since this study focusses on snow on

ground but not snow on glaciers, the few pixels with glacier surfaces were removed in the post-processing in order to reduce the uncertainty of our results.

The above-described meteorological data of the stations were spatially interpolated to the grid of the DEM by inverse distance weighting (IDW) except the radiation components, which are physically calculated in the energy balance module. Vertical gradients were calculated based on the data of the AWS for an hourly time step by IDW. If the correlation coefficient of the vertical gradient is lower than 0.7, Alpine3D omits the data of the AWS with the worst fit to the linear regression. This process is iterated until the correlation coefficient becomes higher than 0.7. If this threshold is not reached, a constant value (independent of altitude) is assumed. For details see Bavay and Egger (2014). Finally, the different snow pack parameters were calculated for each pixel of the DEM grid based on the interpolated meteorological input parameter. The different soil and vegetation were also considered as well as the boundary layer parameters. For example, the roughness length was set to 7 mm and the height of the wind measurement was set to 3.5 m. The condition of the boundary layer was assumed to be neutral. The surface heat fluxes were parameterized using the Monin-Obukhov similarity theory.

## 2.4 Climate scenarios

Projections of future climate are provided as an extension of the CH2011 climate change initiative. This initiative provides among others daily change values of temperature and precipitation for Switzerland on a 2 km grid (Zubler et al., 2014), which are based on the 20 GCM-RCM model chains of the EU-ENSEMBLES project (Van der Linden and Mitchell, 2009). Three emission scenarios (A2, A1B and RCP3PD) are provided for three different time periods (2020-2049, 2045-2074, 2070-2099), which for convenience are labelled by their middle year 2035, 2060 and 2085, respectively. The three emission scenarios can briefly be described as follows: In the RCP3PD scenario, the only interventional scenario, the emissions are supposed to be halved by the mid of the century and thus the $CO_2$ concentration will be stabilized near 450 ppm at the end of the century (van Vuuren et al., 2011). This scenario limits warming to 1.4 °C in Switzerland (most probable value with respect to the 1980-2009) and likely prevents global warming of more than 2°C since the pre-industrial period. The A1B scenario is characterized by a rapid economic growth with a mixture of fossil and non-fossil energy sources. The maximum population will peak around 2050 and the $CO_2$ concentration is roughly 720 ppm at end of the century. In the A2 scenario a continuously increasing population and a low economic rate of growth is assumed and the $CO_2$ concentration reaches roughly 860 ppm (Nakicenovic and Swart, 2000). Note, this scenario is lower than the high-end scenario RCP 8.5(940 ppm) used in the current IPCC's Fifth Assessment Report (van Vuuren et al., 2011).

The assessment of the changes in temperature and precipitation are based on 20 different GCM-RCM ensemble combinations. The focus of this work is related to the median estimate of these 20 different combinations, which were derived by Bayesian methodology. The upper and lower estimates (extremes) of this dataset, which contains the 97.5 %, respectively 2.5 % quantile of the 20 member ensembles are also considered for some analyses in order to get information about the range of the uncertainties of the temperature and precipitation change. Hereof, it is important to know that this Bayesian methodology contracts in some cases the uncertainty range directly derived from the variation of the original RCM simulations. Therefore, the uncertainty range in this paper should also be seen as indicative only. A simple Delta change approach was used to compile meteorological time series of future Alpine climate. This means that the time series of the

reference period were modified with the provided gridded daily change values of the air temperature (ΔT) and precipitation (ΔP). More information about the calculation of these delta values and about the downscaling and can be found in Zubler et al. (2014). Please note, that the reference period of these delta values (1980-2009) has an overlap of 11 years only with the reference period of the meteorological input data (1999-2012). However, a comparison of the winter temperatures for

example revealed a difference of only 0.06 K between the two reference periods. The gridded delta values were interpolated to the coordinates of the AWS of the two regions. This allows a spatial dependence of the climate change signals, in contrast to a spatial averaging of the deltas over the entire region like in Bavay et al. (2013) and most other studies. The temperature change signal can differ by up to 0.5 degrees in the same region (mainly dependent on altitude) and has to be considered. Finally, the ΔT is added to the time series of the air temperature and the ΔP is multiplied by the time series of the

precipitation for each year of the reference period by a simple delta approach.

$$T_{scen} = T_{ref} + \Delta T \qquad (1)$$
$$P_{scen} = P_{ref} \cdot (1 + \Delta P) \qquad (2),$$

where ΔP is given in [%]. A time series of ΔT and ΔP shows the seasonal variations of the climate signals. The highest median estimate of ΔT was clearly found for summer season. The lowest ΔT is predicted for spring, with only slightly higher changes in winter and autumn. Depending on the emission scenarios, the seasonal range of the median estimate of ΔT varies for the end of the century between 0.4 and 1.0 degree. In contrast to temperature, the change in the seasonal precipitation can only be predicted with high uncertainties. The projection range of the 20 different climate models is covering decreasing and

increasing precipitation for almost all seasons and scenario periods. The median estimate of precipitation changes significantly only in summer time, with a decrease of up to 30% towards the end of the century. In spring the precipitation is predicted to increase by up to 10%. For winter and autumn the precipitation will not change significantly. Investigating snow related questions, the climate change signals for winter and beginning of spring are more important than changes in summer time.

In Figure 2 the median estimate deltas and their uncertainty range are shown for the means of the months January to March. Slightly higher temperature changes in Grisons than in the Aare region are projected, especially for the end of the century. In this scenario period the precipitation increases according to the A2 scenario by 4.3% in the Aare region and 7.3% in Grisons, respectively. The influence of the precipitation change is negligible compared to the temperature changes, because the predicted changes in precipitation are very small in the winter half year (Schmucki et al., 2015a).

Due to the fact that the parameterized incoming longwave radiation (ILWR) is a function of the temperature, we calculated the parameterization of the ILWR for each emission scenario separately. This implies an emission scenario dependent ILWR, which is necessary because the fluxes of the ILWR contribute significantly to the snow melt, especially in spring (Schlögl et al., 2016).

Changes of glacier coverage were provided by Linsbauer et al. (2013) and used in order to adapt the land cover data to future scenarios. The changes were calculated with an elevation dependent ice thickness model (M2) for the three emission scenarios and the three different time periods. Future glacier free areas were assumed as pixels with rocks in the land cover data. Note that the ice thickness model is only forced by the temperature change. Changes in the precipitation as seen in

Figure 2 were neglected in the model because of too high uncertainties in the assessment of future precipitation. The current relative amount of glacial areas in the Aare region (6.7%) is higher than in Grisons (1.7%). The future glacier covered area will be halved until 2060 and only a few pixels will still be covered with glaciers towards the end of the century.

## 3 Results and Discussion

We present projected changes of snow depth and duration for two Alpine regions based on the difference between the simulated values of the reference period and 9 different climate projections (3 time periods and 3 emissions scenarios). The results are mainly based on the median estimate of all 20 model combinations, however in the last paragraph the uncertainty based on the 95% spread (upper and lower estimates) is also shown. We often show results for both Alpine regions, but sometimes we focus on the Aare region only since the results are quite similar and its area below 500 m is larger and more

homogeneous than the corresponding elevation zone in the Grisons region.

### 3.1 Validation

By comparing the modelled Alpine3D snow depths of the reference period with measured snow depths, the model fidelity is estimated by means of the RMSE. The nearest and all neighbouring pixels (totally nine) were considered for comparison with the station value. The pixel which showed the best agreement with the station elevation was chosen for comparison.

The agreement is generally good, but such a comparison in a heterogeneous topography like the Alps will always be limited by the fact that the observations are point measurements in a flat field and the pixel value represents an average over an area which is inclined and on a different elevation. Moreover, measured snow depth in high-alpine flat fields usually is higher than the spatially averaged snow depth, e.g. from a grid cell (Grünewald and Lehning, 2015) and therefore generally not representative of a larger area. The RMSE was calculated for each of the 13 years of the reference period for the observed

snow depth above 0.01 m (Table S1). Mountain stations generally show a higher RMSE due to above described topographical effect. Figure S1 illustrates some typical cases, where the simulated snow cover is either too large or too small: At the high elevated station Weissfluhjoch (2540 m) in the Grison region the simulation underestimates the snow depth, whereas several stations between 1000 m and 1700 m asl, especially Disentis (1190 m) tend to start melting snow later than observed. This is also the reason why the simulated snow depth was overestimated for the lowland station Bern

(542 m). This could be partly caused by a known limitation of the albedo function as described in Schmucki et al. (2015a). High RMSE values at high-alpine sites are also explained by the fact that the measured precipitation is often heavily affected

by the uncertainty of the under-catch correction and often shows poor elevation dependence because the regression is sometimes calculated across mountains ranges with different climate on each side. A regression across smaller areas or across the same climate regions (as in the Aare region) would probably improve the linear regression (Schlögl et al., 2016). A comparison with the station based approach of Schmucki et al. (2015a), which also used some of the stations in our investigated regions, demonstrates that the error in simulating the mean winter snow depth in the reference period at the point scale (between -15 and 26 % for the different stations) increases to a larger range by our regional approach (between -47 and 65 %). This is not surprising taking into account the uncertainty involved by the pixel approach in a highly complex mountain area. However, the median value of all stations reveals a comparable uncertainty for both approaches (9 vs 11 %).

## 3.2 Mean snow depth

The impact on the mean snow depth has been investigated by computing the temporal (13 water years) and spatial (area of the region) mean relative changes for the nine different climate projections. The calculated mean snow depth can also be seen as a proxy for the mean snow volume. The analysis reveals similar relative decreases for the two different regions (Figure 3). The Grisons region however always shows somewhat larger decreases, which can be explained by the slightly higher ΔT in the winter months. Concerning the median estimates, the influence of the precipitation change is negligible compared to the temperature changes, because the projected changes are small in the winter half year. The A1B and A2 simulations show similar snow depth changes for the first two scenario periods (ca. -25 %, respectively ca. -50 %). The last scenario period however reveals a ca. 7 % larger decrease for the A2 scenario (ca. -70 %). The RCP3PD simulation however reaches its maximum impact (ca. 30 % decrease) already in the second scenario period. The exact numbers of the median estimate relative decreases are represented by the X-values in Figure S2, where the distribution of the annual changes is also illustrated. To get a better perspective on what these numbers imply for a typical mid-winter day, we visualized the impact for the Feb 1 snow depth (Figure S3) for the end of the century. The greenish colours in the lowest elevations clearly illustrate the strong snow reductions (80-100%) in the regions where most of the population lives. Since the numbers of the relative decrease for snow depth and SWE are very similar (Schmucki et al., 2015a) our results can at least for the Grison region and the A1B scenario be compared with the numbers found in Bavay et al. (2013). Their SWE reduction of 11 to 28 % (depending on the RCM being used) for 2035 and 43 to 66 % for 2085 is in good agreement with our median estimate value of ca. 25 % for 2035 und 62 % for 2085.

## 3.3 Inter-annual variability

The Delta change method applies changes in temperature and precipitation, which depend only on time period and emission scenario but are otherwise constant. Therefore changes in future climate variability, which may be present in the original RCM model predictions, are neglected. According to climate model projections there are no clear signs how future temperature and precipitation variability will evolve in winter in midlatitudes (Deser et al., 2012), although a recent study

indicates a slight decrease of winter temperature variability (Holmes et al., 2016). The analyzed inter-annual variability in this study is therefore first of all determined by the inter-annual variability of the underlying temperature and precipitation conditions in the reference period. For the future scenario periods the shown inter-annual snow variability is additionally influenced by the non-linear dependence of snow on temperature, which changes the variability dependent on the size of the

ΔT values. We characterized the inter-annual variability of the mean snow depth by the d-value (Figure S2). The d-value defines the difference between the year with the highest and the year with the lowest relative decrease for each scenario period and emission scenario. For a very cold winter the relative snow depth decrease will be lower than for a warm winter due to the 0°C dependence. The resulting inter-annual variability in the Grisons region amounts to ca. 15 % in the first scenario period and up to 30 % towards the end of the century. The reason for this increase in d-values is caused by the fact

that the first scenario period is located closer to the reference period than the third and warmest time period. Thus a higher climate change signals usually broadens the distribution. This is only true if there is still enough snow available for melt. This is no longer the case at the end of the century in the 400 m lower-elevated Aare region, which therefore shows no increase in the d-value for the last scenario period. This is also the reason for the slightly higher d-values in Grisons for all emission scenarios in the last two scenario periods.

We also analysed the mean snow depth evolution (mean of 13 years of simulation) and its variability (minimum and maximum snow depth for each day) in six elevation zones for the reference and the last scenario period for the A2 simulation (Figure 4). The snow depth maxima at the end of the century are lower than today's mean snow depth in all elevation zones, except the highest (3000 m asl), where the maxima correspond more or less to today's mean values. On the other end, the mean snow depth evolution at the end of the century is similar to today's minima for all elevation zones except

the highest.

As mentioned earlier the future snow depth is mainly dependent on the increasing winter temperature since the precipitation change in the winter half year is small. The evolution of the mean winter temperature and the maximum snow depth therefore are correlated (Figure S4). At the 500 m (450-550m) elevation zone 6 (3,0) out of the 13 years in 2035 (2060, 2085) show a higher maximum snow depth than the lowest maximum in the reference period. The same figure also reveals

that in the elevation zones between 500 and 1500 m asl the winter with the lowest maximum snow depth in the reference period corresponds to about the winter with the highest maximum snow depth at the end of the century. At the 2500 m, 12 (11,8) out of 13 years show a higher maximum snow depth in 2035 (2060, 2085) than the lowest maximum of the reference period. At these higher elevations more winters remain with maximum snow depths higher than the current minimal snow depth. This is caused by the fact that the colder baseline climate makes snow conditions at higher elevations less sensitive to

warming. . The same results are also valid for the mean snow depth (not shown).

### 3.4 Seasonal and elevation dependence

The relative decrease of the snow depth is dependent on time and elevation zone (Figure 5). The highest relative decrease can be found in the lower elevations. Below 1000 m asl the relative decrease is more than 70% for all emission scenarios and

time periods. Elevations above 2000 m asl are less sensitive to climate change. Nevertheless, even at 3000 m the snow depths will be halved towards the end of the century according to the A2 scenario (Table 1). This in good agreement with a study of Rousselot (2012) in the French Alps, which found a 69 % decrease at 1800 m asl (compared to 75% in our study). The graphs for the RCP3PD scenario demonstrate that the benefit of interventions is only discernible after the first scenario period and then mainly above 2000 m, where the snow cover reduction is limited to about 20%. The begin and the end of the snow season are more sensitive to climate change due to generally warmer temperatures than the mid-winter months January and February, which is especially obvious in higher relative decreases in the spring months (Table 1). This finding is in agreement with Steger et al. (2013), which also observed the largest snow cover reduction in spring.

Due to the fact that daily mean and maximum snow depths are decreasing, the total volume of snow must also shrink. In contrast to the relative decrease the absolute decrease is small below 1000 m asl for the end of the century, since the usual snow volume is small anyway in this elevation zone (Figure 6). The absolute decrease is largest between 1500 and 2500 m asl, since this elevation band is nowadays always snow covered during the winter months and heavily affected by warmer temperatures. This is not the case above 3000 m asl, where absolute decreases are again small since it is usually still cold enough to prevent melting in the winter months. This is also true for the inter-annual variability of the January to March period (shaded areas in Figure 6), which clearly decreases with increasing elevation due to the fact that at higher elevation the snow volume is mainly dependent on precipitation and much less on temperature.

### 3.5 Continuous snow cover

The date of the first continuous snow (snow depth at least 1 cm) and the end of the snow season was calculated based on the longest snow covered period for each of the 13 years for all time periods. Finally, the median of these 13 years was calculated for 100 m elevation bands. The results of this process for the Aare region and the A2 emission scenario are shown in Figure 7. At 1500 m, for example, the snow season starts on average about 2 (2035) to 5 (2085) weeks later and ends 2 (2035) to 12 (2085) weeks earlier. The retreat of the snow disappearance is also dependent on elevation, especially for the end of the century, when the most sensitive elevation zone is at roughly 1500 m asl. This is probably caused by the fact that this elevation zone is then closest to the 0°C limit. At higher reaches, many parts of the winter will still remain below the freezing level. At lower reaches the snow season is anyway too short to produce important reductions in the period length of continuous snow cover. These results confirm the finding of Kotlarski et al. (2015), who investigated the elevation dependency of the number of snow days in 5 RCMs and found a maximum reduction for the winter half year at about1500 m asl.

The snow season at 1000 m asl currently lasts about 4 months from December until end of March. At the end of the century almost no snow is projected at this elevation. A similar reduction of 4.5 months can be observed at 1500 m asl, where the continuous snow cover is reduced to only 2 months, i.e. mid of December to mid of February. Please keep in mind that these numbers are based on an average winter in the corresponding time period and neglect the fact that future winters in this elevation will often be characterized by ephemeral snow cover, which is nowadays typical for elevation below 1000 m only.

This result is in good agreement with the findings of Schmucki et al. (2015b), who demonstrate that at 1500 m asl in the Swiss Alps the probability for a winter with a continuous snow cover is only 60% at the end of the century. Generally, the decrease in snow duration is equal to an elevation shift of 200-500 m for the first scenario period and 700-1000 m for the last scenario period for the A2 scenario. This result is in agreement with a study of Bavay et al. (2013), who used 3 RCMs only and found similar numbers for the Swiss Alps. The little bump at 2800 m in the curve of the first scenario period (Figure 7) results from the smaller glacier coverage in the time period 2020-2049. Originally deleted pixels (due to glacier coverage) are now snow covered pixels in this time period (see 2.3).

### 3.6 Number of snow days

The demonstrated decrease in snow depth and snow duration affects also the number of snow days. We define a snow day as a day with a least 5 cm snow on the ground, because with respect to winter tourism this is the minimum snow depth to generate a winter feeling, to build a snow man, or to go sledding. The number of such snow days was therefore calculated for the four time periods for several towns in the two investigated regions. Table S2 shows the median number of such snow days for the A2 scenario. The results clearly show that the number of snow days in the Swiss plateau will become zero in the last scenario period. Therefore, a multi-day snow cover will be a rare event towards the end of the century in this elevation zone. Stations at about 1500 m will lose ca. 100 snow days, especially in the melting season. Davos (1560 m), for example, is supposed to have only 10 snow days more at the end of the century than Chur (593 m) today and Adelboden (1350 m) will get less snow days than Bern (542 m) at the present time.

The inter-annual variability in snow days is shown in Figure 8 for three selected stations in the Aare region. The range is highest for Bern (542 m) at the present time, for Grindelwald (1034 m) in the first scenario period and for Mürren (1650 m) in the last scenario period, which corresponds well with the findings described in section 3.3, where the elevation with the highest variability increased with time. Note that the inter-model variability, from which the median estimate is calculated, is much smaller than the inter-annual variability as shown in Schmucki et al. (2015b).

The probability of a winter with 0 snow days, less than 5, 15 or 50 snow days depending on elevation and scenario period is shown Figure S5. As expected, to the same probability in future would be found at higher elevation. For example, today there is a 7% probability that we experience less than 5 snow days at 500 m asl. During the middle of the century and the A2 emission scenario the same probability can be found at 850 m asl.

A higher snow day threshold has to be taken into account, when the natural snow reliability for a ski resort is analyzed. The snow reliability is an important factor for a profitable ski resort and is directly correlated with the expected costs for the additional production of artificial snow. A minimum snow depth of 30 cm during 100 days between December 1 and April 15 is often used a threshold for this purpose (Elsasser and Bürki, 2002), because experiences show that this is the minimum requirement for an economically viable ski area operation. To illustrate the declining elevation- and time-dependent natural snow reliability the median number of days where at least 80 % of the pixels have a snow depth of at least 30 cm were

therefore calculated in 200 meter elevation bands. Figure 9 demonstrates the thus calculated current and future snow reliability for the A2 scenario in the Aare region separated by north and south facing aspects. Elevations and time periods with less than 40 days with at least 30 cm snow on the ground are colored in red. Green colors in contrast are indicating a snow guarantee for the ski resorts (more than 100 days with at least 30 cm snow). The cases between 40 and 100 days are

labelled yellow, which indicates elevations and time periods, where the natural snow reliability is marginal and local effects may be a dominating factor.

According to this approach the natural snow cover is already today definitely not sufficient below 1000 m. This elevation limit is shifted to 1800 (2000) m asl at the end of the century for north (south) facing slopes. On the other end, today enough snow can only be guaranteed above 1400 (1600) m asl for north (south) facing slopes. In 2085 however a natural snow

guarantee can only be found above 2400 (2600) m asl for northern (southern) aspects. This upward move of the snow reliability of 800-1000 m between the reference period and the last scenario period is within the elevation shift range found for the continuous snow cover (section 3.5). Compared to observations the 200 m elevation difference in snow reliability between northern and southern slopes seems to be on the low side but can be explained by the following fact. Even the high-resolution 200 m DEM produces a smoother topography, in which slopes are less steep and therefore southern aspects less

exposed to the low winter sun. In addition, small scale processes due to rough terrain (e.g. enhanced melting in rocky slopes of southern aspect, drifting snow) are not considered in our modelling set-up.

Snow is not only an important economic parameter for winter tourism, snow also plays in important role for the evolution of the permafrost in high Alpine regions. This permafrost will probably thaw in a warmer climate if it is not protected by a deep and long-lasting snow pack in spring and summer (Haberkorn et al., 2015). The retreat of the permafrost in the alpine

regions can affect the stability of infrastructure in these area or cause debris flows, which threaten populated areas far downstream (Haeberli et al., 2010). For this purpose we analyze the number of days with at least 30/50 cm of snow at 3000 m asl in all four time periods for the A2 scenario in the Aare region. There are currently 308 such snow days with at least 50 cm snow at this elevation (Table 2). The reduction is 30 % in the first scenario period, 40 % in the second and 60% in the last one, leaving only 133 snow days. Note that this is a much higher reduction than the corresponding decrease in snow

cover duration at the same elevation, which only accounts to ca. 30 % in the last scenario period (Figure 7).

### 3.7 Uncertainty consideration

Ideally the uncertainty analysis incorporates every step in the modelling process. The biggest uncertainty has been assessed by considering three different emission scenarios. Another source of uncertainty comes from the snow model in regard to the resolution, the parametrized processes, the choice of the boundary layer parameters and the available meteorological stations

to verify the RCM runs. Schlögl et al. (2016) concluded in a recent study that the uncertainty of the simulated snow water equivalent  from these factors is typically ca. 15%, but is negligible in climate change studies as long as only relative changes are considered. As described in Zubler et al. (2014) there is also uncertainty in the RCM downscaling procedure. One important point to keep in mind is the interpretation of the high elevation results, because the highest point in the

ENSEMBLES grid is only 2600 m asl. Moreover, above this elevation glacier melt may decrease the spring snow albedo by dust from exposed moraine rubble and glacial till (Oerlemans et al., 2009).

In the following we focus on the uncertainty originating from the different temperature and precipitation changes as projected by the 20 different GCM-RCM chains available from the CH2011 initiative. So far the focus was on projected changes based on the median value from the ensemble of these different models. To investigate this uncertainty the snow cover has also been simulated for the upper and lower $\Delta T$, respectively upper and lower $\Delta P$ (see 2.4), for the three future time periods and the A2 emission in the Aare region. These four simulations define the range of the uncertainties of the climate change signal.

– upper $\Delta T$, upper $\Delta P$
– lower $\Delta T$, lower $\Delta P$
– upper $\Delta T$, lower $\Delta P$
– lower $\Delta T$, upper $\Delta P$

Simulating only the first two cases is not sufficient because the upper $\Delta T$ does not necessarily imply an upper $\Delta P$ and a lower $\Delta T$ does not necessarily imply a lower $\Delta P$ (Fischer et al., 2012). The correlation between $\Delta T$ and $\Delta P$ depends on season, region and the future time period and varies from a negative inter-variable relation (lower $\Delta T$ and upper $\Delta P$, respectively upper $\Delta T$ and lower $\Delta P$) mostly in the summer time to a positive inter-variable relation (lower $\Delta T$ and lower $\Delta P$, respectively upper $\Delta T$ and upper $\Delta P$) in winter time.

Not surprisingly, in each scenario period the highest relative decrease was found for the upper $\Delta T$ & lower $\Delta P$ simulation, whereas the lowest relative decrease was found for the lower $\Delta T$ & upper $\Delta P$ simulation (Table 3 & Figure 3). These lower and upper estimates have been calculated for the Aare region for the A2 scenario only due to limitations in computational power. The spread between the lowest and highest estimate is quite high. With the exception of the scenario for 2035 (increase of 13 %) even the lowest estimates cause mean snow depth decreases between 13 and 34 %. The highest estimates in contrast cause decreases between 49 and 84 %.

Figure S6 demonstrates the effect of these possible combinations on snow depth dependent on elevation for all three scenario periods: The upper panels show the impact on the absolute snow depth and the difference relative to the mean changes as considered in the above paragraphs. The lower panels illustrate the relative difference in snow depth between the upper and lower $\Delta T$ simulations with the lower $\Delta P$ configuration (T_low) and with the upper $\Delta P$ configuration (T_high), respectively the relative difference in snow depth between the upper and lower $\Delta P$ simulations with the lower $\Delta T$ configuration (P_low) and with the upper $\Delta T$ configuration (P_high).

The difference in snow depth between the two extreme precipitation scenarios is largest at about 3300 m asl for the P_high and the P_low simulation in the 2060 and 2085 scenario period, indicating that these elevation zone is most sensitive to a change in the precipitation amount. The first scenario period shows no clear peak for the P_low simulation because changes

in precipitation at a small temperature increase affect all elevation above about 1000 m asl similarly. As expected, the difference is larger at with lower temperature configuration than with the upper temperature configuration in all three scenario periods.

On the other end, the difference in snow depth between the two extreme temperature scenarios peaks at about 1000 m asl for the 2035 scenario period indicating that these elevation zone is most sensitive to temperature changes in the near future. The 2060 and 2085 scenario periods show a peak at 1800 m, resp. 2300 m asl demonstrating that the most sensitive elevation zone is increasing with time independently of the precipitation change (T_high or T_low). However, the difference is larger with the upper precipitation configuration than with the lower precipitation configuration in all three scenario periods.

## 4 Conclusions

The large set of downscaled climate models used in this study demonstrates a clear temperature increase for all time periods and emission scenarios. The precipitation signal on the other hand is diverging between the individual models. Future seasonal temperature increase is projected to be highest in summer and lowest in spring with only slightly higher changes in winter and autumn. This is in contrast to the observed temperature increases during the last decades, which are largest in spring, closely followed by summer. Median estimate values of these projected changes together with a measured climatology (reference period) were used as input for Alpine3D to analyse the impact of these changes on the future snow cover for two different Alpine regions. Our results corroborate the general findings of earlier studies, but quantify the uncertainty much better because the median estimate and the lower and upper bounds (2.5 and 97.5 % quantile) from 20 different GCM-RCMs were used as input. Moreover, in addition to the often used A1B and A2 emission scenario, the benefit of an intervention scenario (RCP3PD) was investigated, which allowed analyzing how much Alpine snow can be saved if we manage to stabilize the global temperature increase below 2°C relative to the preindustrial level.

The results demonstrate that the duration and mass of the snow cover in typical Alpine catchments such as the Aare and Grisons region will shrink until the end of the century independently of the emission scenario and the used climate model. However, the size of the decrease can be heavily reduced with an intervention scenario. Both regions show a similar clear reduction in the future snow volume (Jan-Mar) based on the median estimate values. For the A1B emission scenario the expected snow volume reduction averaged over both region will be about 25 % in the near future (2035), 50 % towards the middle of the century (2065) and 60 % towards the end of the century (2085). The higher A2 scenario differs only at the end of the century, where the reduction increases to ca. 70%. The RCP3PD scenario however, can limit the expected snow reduction to 30 % after the middle of the century.

Since the current emissions do not follow the RCP3PD track (Peters et al., 2013), the following paragraph refers to the A2 scenario: The expected snow volume reduction increases from ca. 50 % above 3000 m to almost 100 % at lowest elevations (500 m asl). Similarly, the reduction increases e.g. at 1500 m asl and for mid of the century from ca. 50 % in mid-winter to almost 100 % in spring. A detailed analysis of the inter-annual variability demonstrates that the snow cover of a nowadays

snow-scarce winter below 1500 m can be expected to be the average snow cover at the end of the century. The elevation of the largest absolute declines is increasing with time from 1000 m asl during the first scenario period (2035) to 1800 m asl during the second period (2065) to 2300 m asl during the third period (2085), because the elevation with the conditions for melt and the maximum available snow are also increasing with time. It is obvious, that such reductions in snow volume also imply a decrease in snow duration. Our analysis reveals that the snow duration at 2000 m decreases by 2 weeks in 2035 and by 11 weeks in 2085. Thus, already in mid of the century at low elevations between 500 und 1000 m asl there will be only a few days with snow cover left. The generally shorter and thinner future snow cover is equivalent to an elevation shift of ca. 400 m in 2035 and ca. 900 m in the last scenario period. These numbers demonstrate that projected snow reduction is highly dependent on elevation and season. Considering also the lower and upper bounds of the projections reveals that the snow volume reduction has an uncertainty of about ±25%, which implies that Alpine snow may decrease by at least 30 % even under the most favourable conditions (low temperature increase, high precipitation increase) until the end of the century.

The clear decrease in future snow depth and snow duration as shown above negatively affects the society by decreasing natural snow reliability for ski resorts and by decreasing the protection offered by the snow cover in high altitude permafrost areas. Since the snow cover of the two investigated regions finally ends as melt water in three of the major rivers in Central Europe (Rhine, Danube and Po) the changing seasonal runoff might heavily impact the water usage downstream (e.g. hydropower, irrigation, transportation), especially during the projected dryer summer months (Beniston and Stoffel, 2014). Furthermore, our results clearly demonstrate that at low elevations, where the majority of the population in the Alpine area lives, a multi-day snow cover will become a rare event after mid of the century. In these heavily populated areas the vanishing snow cover due to warmer winter temperatures may also have positive side effects on the society because a decreasing number of frost and snow days are positively correlated with the number of road accidents (Norrman et al., 2000), airport closures (Hess et al., 2009), traffic interruptions as well as the cost for winter road maintenance (Schmidlin, 1993).

We want to reemphasize 1) that the projections are based on the Delta change approach, which implies that the variability does not change over time and 2) that the presented results are mostly based on the expected median estimate changes and on a climatologically averaged snow cover. The uncertainty analysis demonstrates that the range of uncertainty in the simulated snow cover decrease is determined by the inter-annual variability and the uncertainties in the climate change signal of the different RCM projections. Due to the fact that precipitation towards the end of the century is projected to slightly increase, it is therefore probable, that also in future (then) unusual winters may experience short periods of abundant snow, which might have worse consequences than today since society probably would be less prepared for such rare events.

**Acknowledgments**

The meteorological input data used for the simulation of the reference period was provided by the Swiss Federal Office of Meteorology and Climatology MeteoSwiss. The snow depth data used for verification purposes was provided by SLF and MeteoSwiss. The climate change signal data were obtained from the Center for Climate Systems Modeling (C2SM) at ETH

Zürich provided by the CH2011 community. We thank Marcia Phillips for proofreading the manuscript. The study was partly financed by a special grant of the Swiss Federal Institute for Forest, Snow and Landscape Research WSL.

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

**Table 1: Relative decrease of snow depth for different elevation zones and months in the Aare region for 2085 A2 emission scenario.**

| Altitude (m) | Oct | Nov | Dec | Jan | Feb | Mar | Apr | May | Jun | Oct-Jun | DJF |
|---|---|---|---|---|---|---|---|---|---|---|---|
| < 500 | 1.00 | 1.00 | 0.97 | 0.93 | 0.94 | 0.96 | 1.00 | NAN | NAN | 0.97 | 0.95 |
| 500-1000 | 1.00 | 0.96 | 0.96 | 0.95 | 0.91 | 0.94 | 1.00 | 1.00 | 1.00 | 0.96 | 0.94 |
| 1000-1500 | 0.96 | 0.85 | 0.86 | 0.90 | 0.85 | 0.89 | 0.97 | 1.00 | 1.00 | 0.92 | 0.87 |
| 1500-2000 | 0.89 | 0.77 | 0.73 | 0.78 | 0.75 | 0.79 | 0.93 | 0.99 | 1.00 | 0.85 | 0.75 |
| 2000-2500 | 0.78 | 0.66 | 0.63 | 0.63 | 0.57 | 0.59 | 0.75 | 0.98 | 0.99 | 0.73 | 0.61 |
| 2500-3000 | 0.78 | 0.62 | 0.52 | 0.47 | 0.40 | 0.37 | 0.44 | 0.82 | 0.99 | 0.61 | 0.46 |
| 3000-3500 | 0.77 | 0.61 | 0.47 | 0.40 | 0.36 | 0.32 | 0.30 | 0.56 | 0.94 | 0.53 | 0.41 |

**Table 2: Number of days with more than 30 cm/50 cm of snow depth at 3000 m asl in the Aare region for the reference period and the three future scenario periods based on the A2 emission scenario.**

| Snow days at 3000 m | Reference | 2035 | 2060 | 2085 |
|---|---|---|---|---|
| >30 cm | 320 | 247 | 213 | 165 |
| >50 cm | 308 | 220 | 187 | 133 |

**Table 3: Change (%) of snow depth relative to the reference period for the three future time periods based on the A2 emission scenario in the Aare region.**

| 2035 | lower ΔP | upper ΔP | 2060 | lower ΔP | upper ΔP | 2085 | lower ΔP | upper ΔP |
|---|---|---|---|---|---|---|---|---|
| lower ΔT | - 23% | +13 % | lower ΔT | - 40% | - 13 % | lower ΔT | - 57% | - 34 % |
| upper ΔT | - 49 % | - 24 % | upper ΔT | - 68 % | - 52 % | upper ΔT | - 84 % | -.74 % |
| Median ΔTΔP | | - 22 % | Median ΔTΔP | | - 45 % | Median ΔTΔP | | - 66 % |

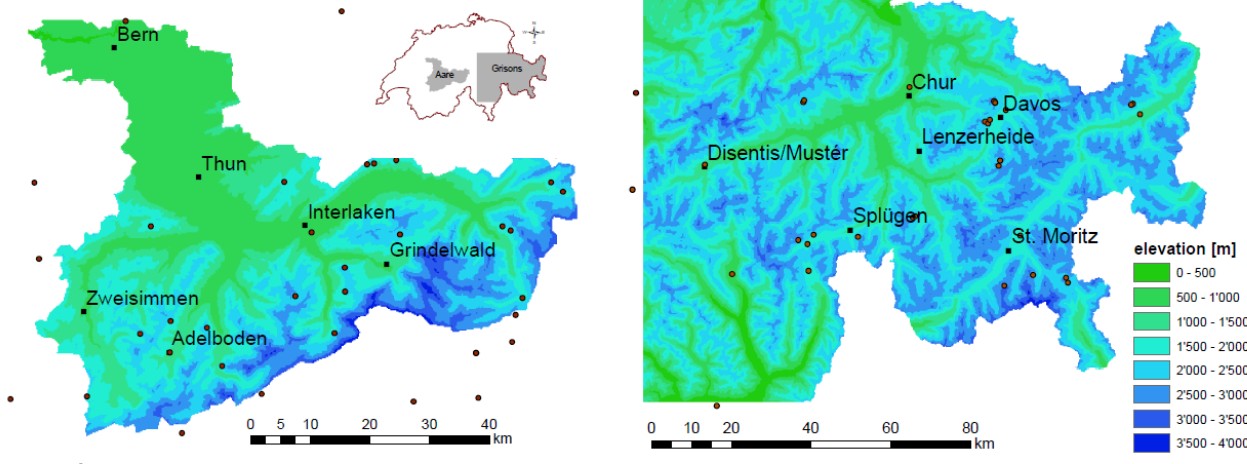

**Figure 1: Elevation of the Aare (left) and the Grisons region (right). Weather stations used for simulations are marked with dots. The location of the two regions within Switzerland is given by the red inset with the two grey areas.**

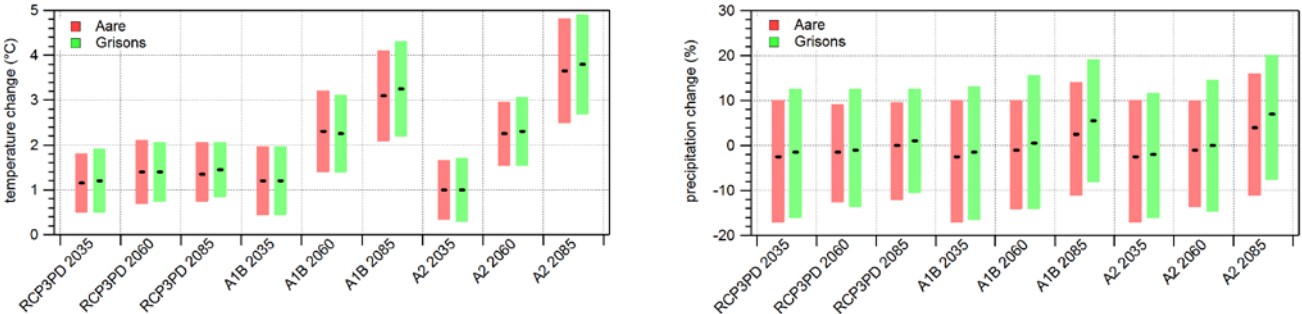

**Figure 2: Mean January to March temperature (left) and precipitation (right) changes including the uncertainty bars from the upper and lower estimates for the Aare (red) and the Grisons (green) region.**

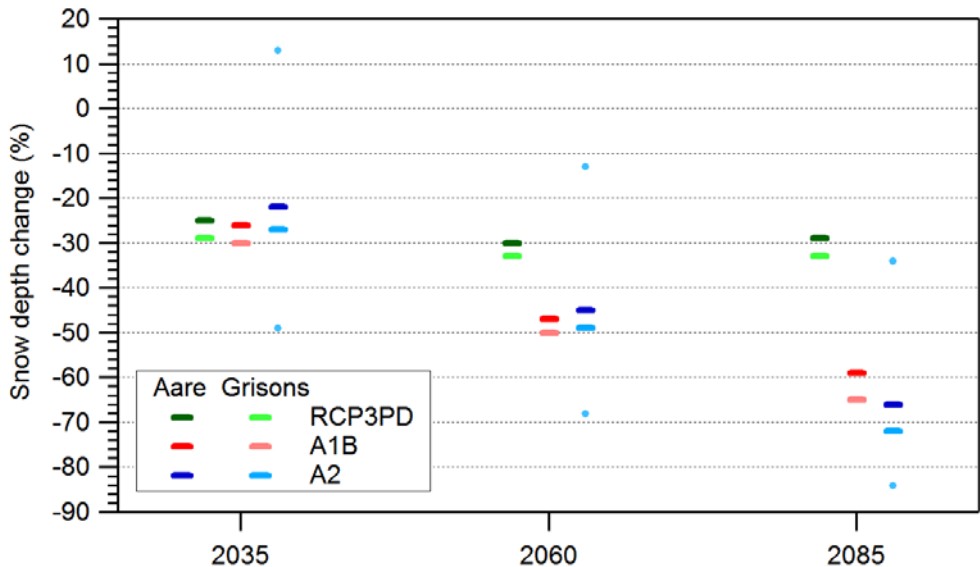

**Figure 3: Decrease of annual mean snow depth (%)relative to the reference period (1999–2012) for the Aare region and the Grisons region for the three different emission scenarios and time periods based on the median estimate change of temperature and precipitation (bars). The lowest and highest estimates (Table 3) are only shown for the Aare regions and A2 scenario (dots).**

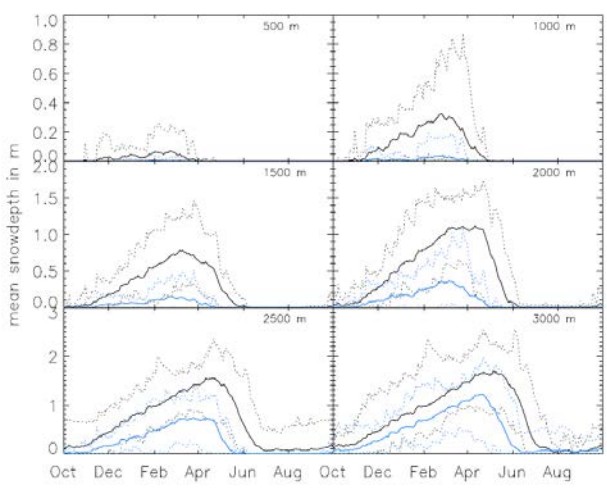

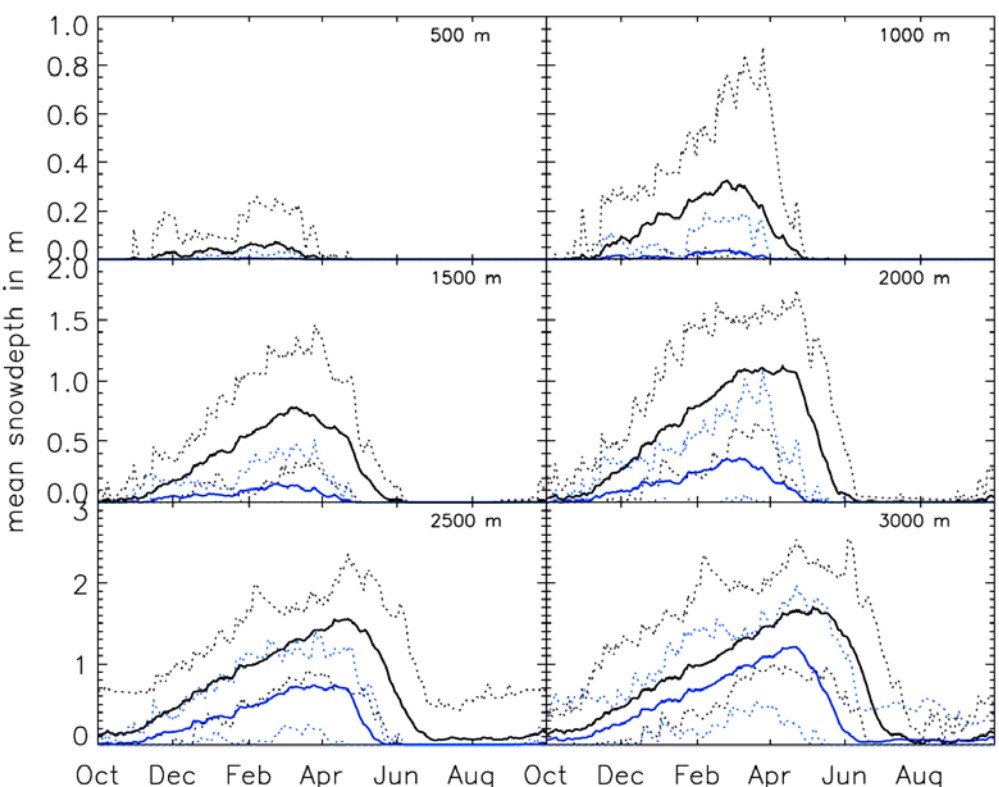

**Figure 4: Mean (solid), maximum and minimum (dotted) snow depth for the reference period (black) and the A2 2085 scenario (blue) in the Aare domain for 6 elevation zones. The elevation zones are 100 m wide, i.e. the 1500 m zone, contains all pixels between 1450 and 1550 m. Note, the scale of the y-axis changes with elevation.**

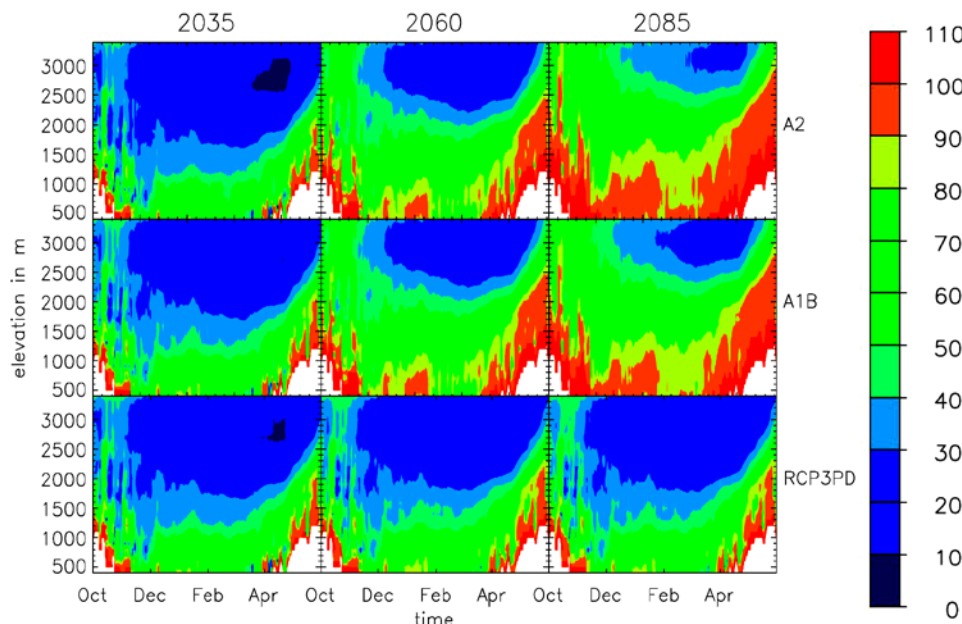

**Figure 5: Relative decrease (%) of the snow depth in the Aare region for the three emission scenarios (top, middle, bottom) and the three different time periods (left to right) dependent on season and elevation. The white colors indicate no data.**

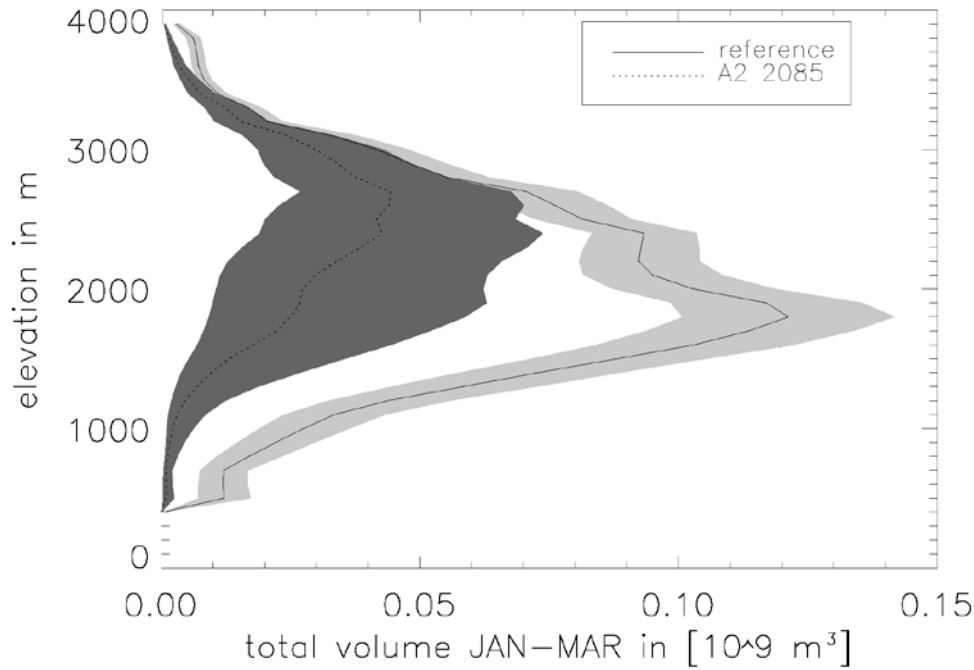

**Figure 6: Total volume of snow (Jan-Mar) in the Aare region for the today (solid line) and the end of the century (dotted line). The shaded area for the reference period indicates half of the standard deviation (for readability) of the inter-annual variability. The shaded area of the 2085 scenario period indicates the range between the lowest and highest estimate based on the A2 emission scenario (Table 3).**

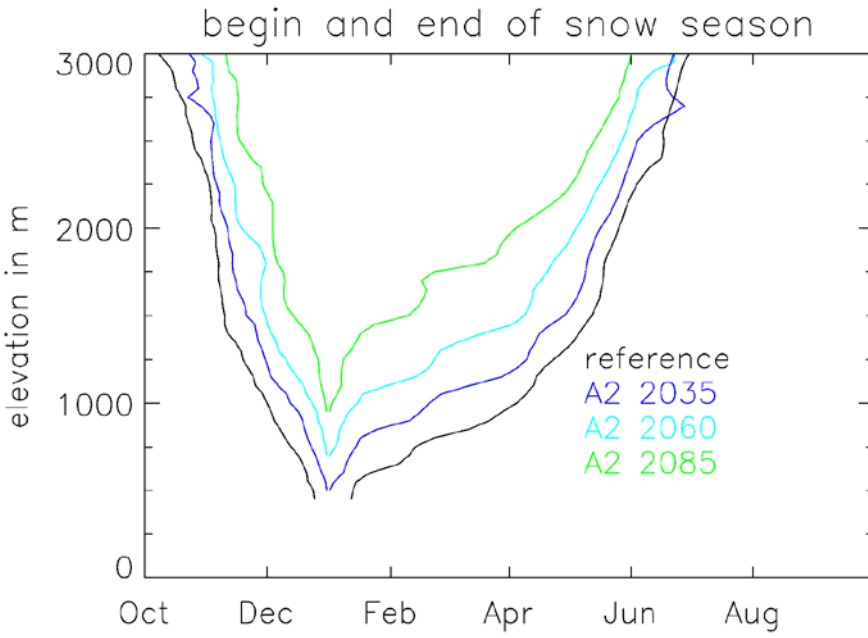

**Figure 7: Begin and end of a continuous snow cover for the A2 emission scenarios for the reference and three future time periods in the Aare region.**

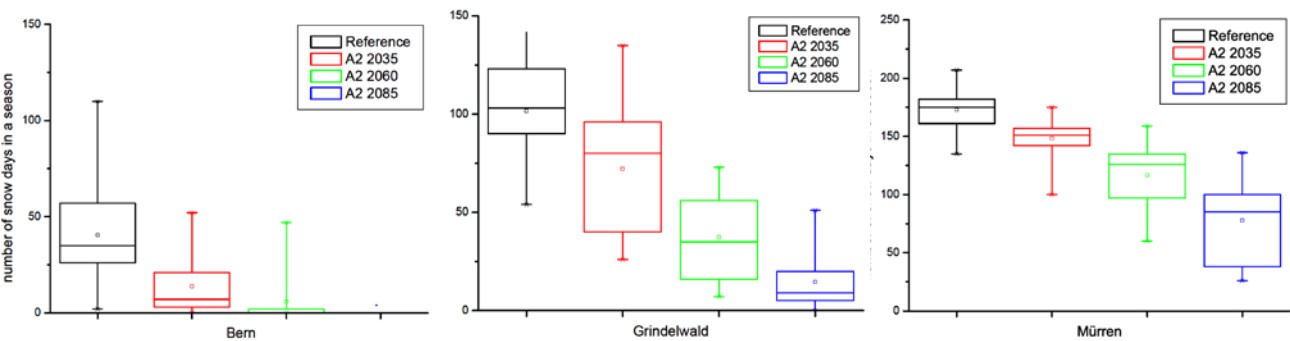

**Figure 8: Inter-annual variability of the number of snow days (snow depth at least 5 cm) at three stations (Bern 540 m (left), Grindelwald 1030 m (center) and Mürren 1650 m (right)) in the Aare region. The little square in the box plots represents the mean value and the whiskers show the 2.5 % and 97.5 % quantile value of the different model simulations.**

**North facing (left):**

| Zone | Reference | A2 2035 | A2 2060 | A2 2085 |
|---|---|---|---|---|
| 600 - 800 m | Red | Red | Red | Red |
| 800 - 1000 m | Red | Red | Red | Red |
| 1000 - 1200 m | Yellow | Red | Red | Red |
| 1200 - 1400 m | Yellow | Yellow | Red | Red |
| 1400 - 1600 m | Green | Yellow | Yellow | Red |
| 1600 - 1800 m | Green | Yellow | Yellow | Red |
| 1800 - 2000 m | Green | Green | Yellow | Yellow |
| 2000 - 2200 m | Green | Green | Green | Yellow |
| 2200 - 2400 m | Green | Green | Green | Yellow |
| 2400 - 2600 m | Green | Green | Green | Green |
| 2600 - 2800 m | Green | Green | Green | Green |
| 2800 - 3000 m | Green | Green | Green | Green |
| 3000 - 3200 m | Green | Green | Green | Green |

**South facing (right):**

| Zone | Reference | A2 2035 | A2 2060 | A2 2085 |
|---|---|---|---|---|
| 600 - 800 m | Red | Red | Red | Red |
| 800 - 1000 m | Red | Red | Red | Red |
| 1000 - 1200 m | Yellow | Red | Red | Red |
| 1200 - 1400 m | Yellow | Yellow | Red | Red |
| 1400 - 1600 m | Yellow | Yellow | Yellow | Red |
| 1600 - 1800 m | Green | Yellow | Yellow | Red |
| 1800 - 2000 m | Green | Yellow | Yellow | Red |
| 2000 - 2200 m | Green | Green | Yellow | Yellow |
| 2200 - 2400 m | Green | Green | Green | Yellow |
| 2400 - 2600 m | Green | Green | Green | Yellow |
| 2600 - 2800 m | Green | Green | Green | Green |
| 2800 - 3000 m | Green | Green | Green | Green |
| 3000 - 3200 m | Green | Green | Green | Green |

**Figure 9: Assessment of the elevation dependent natural snow reliability in the Aare region for north (left) and south (right) facing aspects. Red: Not snow reliable - a minimum snow depth of 30 cm was reached during less than 40 days. Green: Snow reliable - at least 100 days with more than 30 cm of snow depth were observed. Yellow: The cases in between the green and red definition.**

# Electronic Supplement

**Table S1: Root mean squared error (RSME) calculated for snow depth above 0.01 m for selected stations in the Aare and Grisons catchment during the 13 years of the reference period.**

| Stat | Altitude | 2000 | 2001 | 2002 | 2003 | 2004 | 2005 | 2006 | 2007 | 2008 | 2009 | 2010 | 2011 | 2012 |
|------|----------|------|------|------|------|------|------|------|------|------|------|------|------|------|
| Chur | 550 m | 0.04 | 0.03 | 0.03 | 0.03 | 0.03 | 0.03 | 0.03 | 0.06 | 0.03 | 0.03 | 0.03 | 0.03 | 0.03 |
| Bern | 550 m | 0.09 | 0.04 | 0.08 | 0.05 | 0.06 | 0.06 | 0.08 | 0.08 | 0.03 | 0.08 | 0.08 | 0.09 | 0.05 |
| Disentis | 1090 m | 0.18 | 0.17 | 0.18 | 0.18 | 0.16 | 0.17 | 0.18 | 0.19 | 0.17 | 0.17 | 0.18 | 0.15 | 0.18 |
| Arosa | 1320 m | 0.42 | 0.41 | 0.41 | 0.44 | 0.42 | 0.42 | 0.43 | 0.47 | 0.43 | 0.43 | 0.44 | 0.40 | 0.44 |
| Samedan | 1750 m | 0.31 | 0.29 | 0.31 | 0.31 | 0.29 | 0.31 | 0.32 | 0.31 | 0.31 | 0.30 | 0.30 | 0.26 | 0.33 |
| Weissfluhjoch | 2540 m | 0.98 | 0.95 | 1.02 | 1.02 | 0.98 | 1.03 | 1.02 | 1.04 | 1.05 | 1.00 | 0.98 | 1.01 | 1.03 |

**Table S2: Mean number of snow days (snow depth at least 5 cm) for the reference and the three future time periods based on the A2 scenario for 5 stations in the Aare region (above) and the Grisons region (below).**

| Station | Elevation | Reference | 2035 | 2060 | 2085 |
|---|---|---|---|---|---|
| Bern | 542 m | 40 | 13 | 5 | 0 |
| Interlaken | 568 m | 41 | 14 | 6 | 0 |
| Grindelwald | 1034 m | 101 | 72 | 37 | 14 |
| Adelboden | 1350 m | 139 | 107 | 75 | 35 |
| Mürren | 1650 m | 173 | 148 | 116 | 77 |
| | | | | | |
| Chur | 593 m | 39 | 14 | 5 | 0 |
| Disentis | 1130 m | 98 | 68 | 40 | 18 |
| Davos | 1560m | 157 | 123 | 86 | 49 |
| Samedan | 1721 m | 165 | 139 | 101 | 61 |
| Weissfluhjoch | 2690 m | 254 | 228 | 197 | 163 |

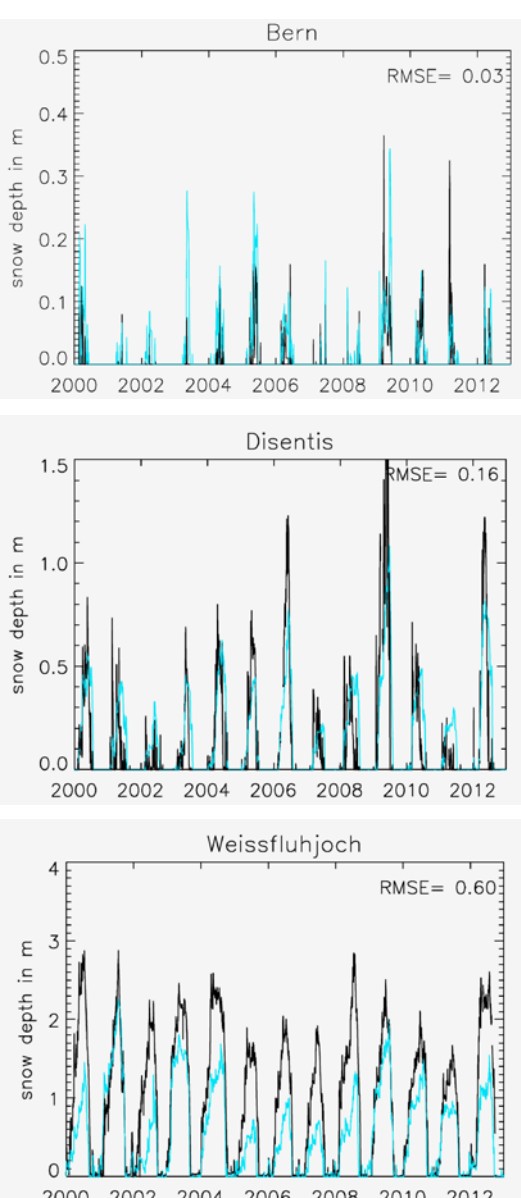

5    **Figure S1: Measured (black) and simulated snow depth (grey) for Bern (542 m), Disentis (1190 m) and Weissfluhjoch (2540 m). Note, the scale of the y-axis is different for each station.**

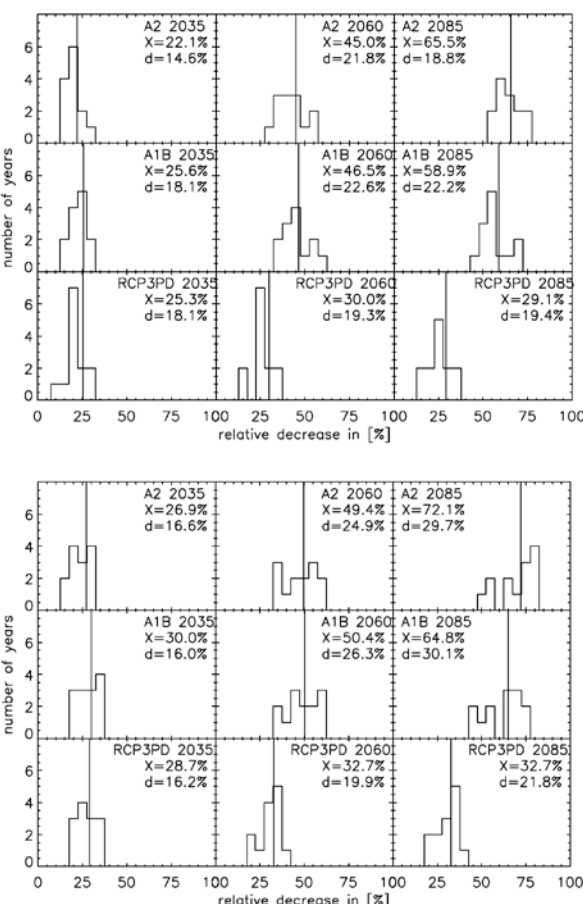

**Figure S2: Distribution of the annual relative decreases of the snow depth for A2, A1B and RCP3PD and the three different future time periods (2020-49, 2045-74, 2070-99) for Aare (top) and Grisons (bottom) based on the inter-annual variability of the reference period. The value X corresponds to the spatial and temporal mean relative decrease in [%], which is visualized in Figure 3. The value d defines the difference between the highest and the lowest relative decrease within the scenario period. The number of the simulated years is N=13.**

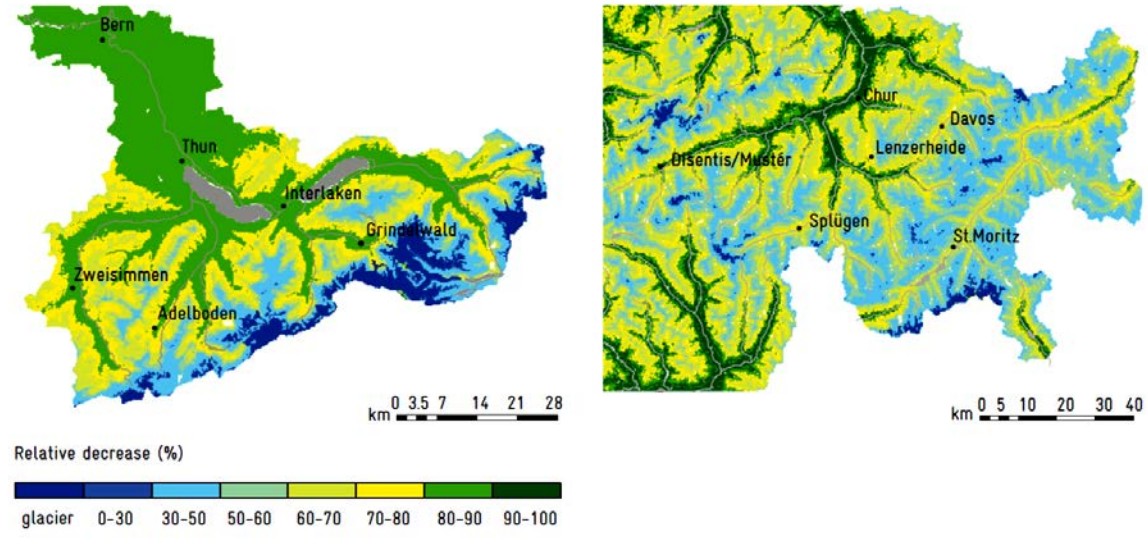

**Figure S3: Relative decrease for February 1 snow depth of the last time period (2085) and the A1B scenario for the Aare region (left) and the Grisons region (right).**

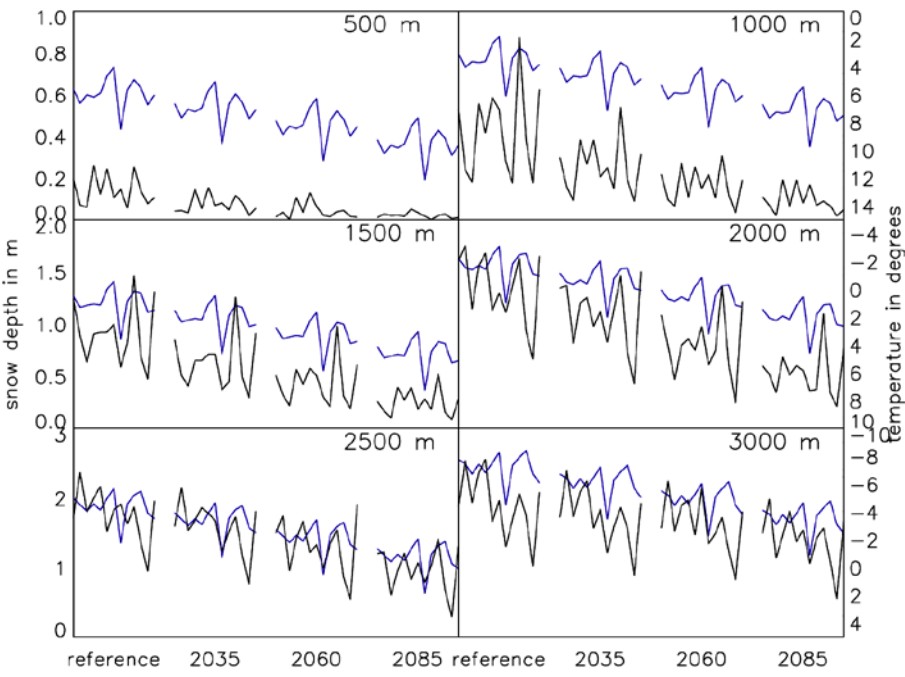

5   **Figure S4: Current and future variability of maximum snow depth (solid line) and mean winter temperature (dashed line) from November to April for 6 elevation zones and the A2 scenario for the Aare region. Note, the scale of the y-axis changes with elevation.**

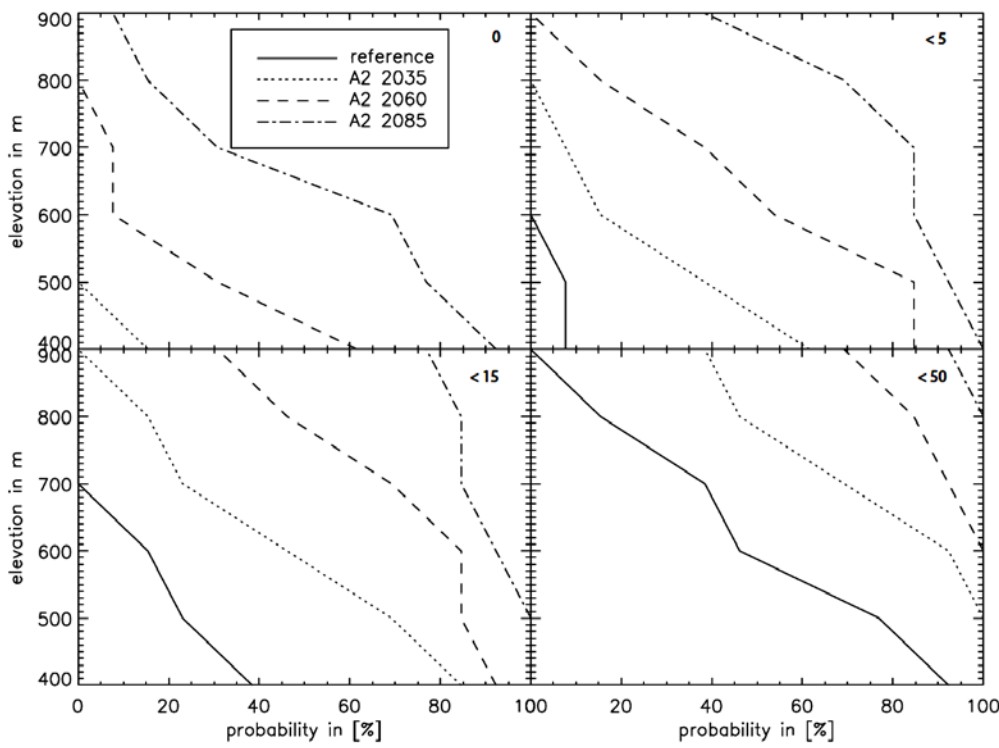

**Figure S5: Probability of winter with 0 (upper left), less than 5 (upper right), less than 15 (lower left) and less than 50 (lower right) snow days for Grisons.**

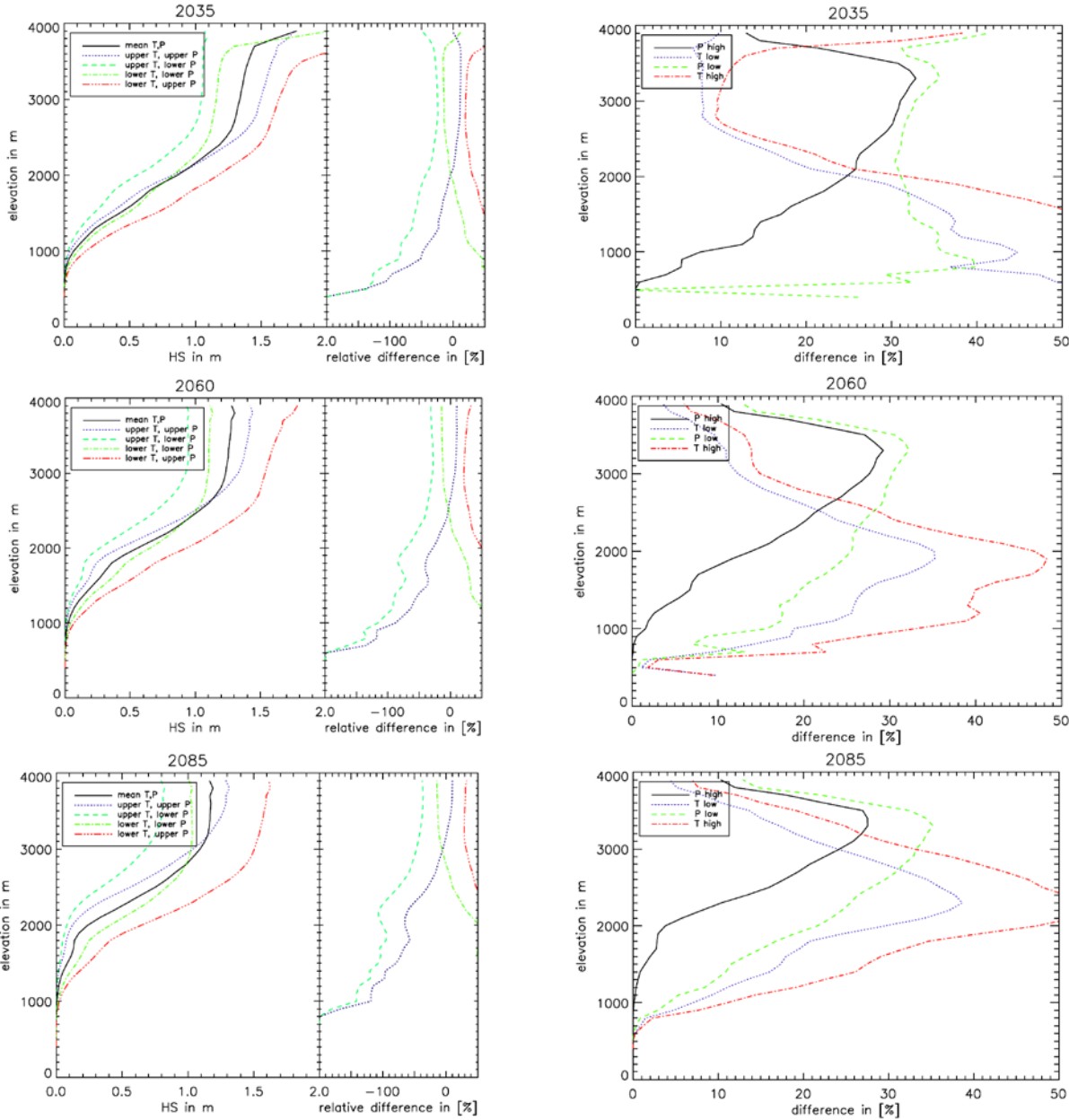

**Figure S6: Elevation dependence of absolute snow depth for five possible T-P combinations (left) and decrease relative to the mean T, mean P configuration (left panels). Elevation dependence of the relative snow depth difference (%) for four different combinations T-P (right panels). "T high" means the difference of the upper T - lower T at the upper P configuration. "P low" means the difference of the upper P - lower P at the lower T configuration. This is shown for 2035 (top), 2060 (middle) and 2085 (bottom) based on the A2 emission scenario in the Aare region.**