# Peer review of "How much can we save? Impact of different emission scenarios on future snow cover in the Alps."

_The Cryosphere, 2016_

## Short Comment (SC1) · 1 Nov 2016

The authors assessed the future projection of snow depth in the Alps by accounting for future temperature and precipitation change under different emission scenarios. The results are interesting and can advance our understanding in the impact of climate change on mountain snow. Here, I have a short comment. Several recent studies (e.g., Painter et al., 2013; Liou et al., 2014; Lee et al., 2016) found that deposition of light-absorbing aerosols (mainly black carbon and dust) substantially decreases snow albedo, which further reduces snow depth and cover. However, this factor has not been considered by the authors in the future projection, which could play an important role. It would be helpful if the authors could include some discussions on these recent findings and the uncertainty due to this aerosol-snow effect in the projection of snow depth.

[Figure]

References:

Lee, W.-L., K. N. Liou, C. He, H.-C. Liang, T.-C. Wang, Q. Li, Z. Liu, and Q. Yue: Impact of absorbing aerosol deposition on snow albedo reduction over the southern Tibetan plateau based on satellite observations, Theoretical and Applied Climatology, 1-10, doi:10.1007/s00704-016-1860-4, 2016.

Liou, K. N., Takano, Y., He, C., Yang, P., Leung, L. R., Gu, Y., and Lee, W. L.: Stochastic parameterization for light absorption by internally mixed BC/dust in snow grains for application to climate models, J. Geophys. Res.-Atmos., 119, 7616–7632, doi:10.1002/2014jd021665, 2014.

Painter, T. H., M. G. Flanner, G. Kaser, B. Marzeion, R. A. VanCuren, and W. Abdalati, End of the Little Ice Age in the Alps forced by industrial black carbon, Proc. Natl. Acad. Sci. U.S.A., 110(38), 15,216–15,221, doi:10.1073/pnas.1302570110, 2013

---

## Referee Comment (RC1) · Anonymous Referee #1 · 14 Nov 2016

The manuscript by Marty et al. is concerned with the assessment of future snow cover changes over two mountainous regions of Switzerland. For this purpose a gridded version of the CH2011 Swiss Climate Scenarios is used to force a distributed model of the surface snowpack. The analysis includes an assessment of projection uncertainties arising from the assumption of different emission scenarios and from climate model uncertainty. In agreement with previous works the study finds an important decrease of future snow cover that considerably depends on elevation and, for the late 21st century, on the choice of emission scenario.

In a general sense, the topic of the work fits very well into the journal's scope and adds a further piece of information to 21st Century climate change impacts in the European Alps. Qualitatively and quantitatively previous works are confirmed employing a new methodology that relies on gridded climate change information and the application of a

spatially distributed snow pack model. As such, I consider the work as being relevant for the scientific community. For most parts, the methods are described appropriately, and the conclusions are well-based on the results obtained. There are only minor language issues. The paper however suffers from a number of inaccuracies in the description of the underlying datasets, from a partly questionable analysis of interannual snow cover variability and from a partly poor figure quality. Please see the listing below for further details. These issues should be improved before publication of the paper. For this purpose, only few new analyses are required and the basic structure of the paper does not have to be changed. I'd therefore suggest to return the manuscript to the authors for minor revisions. I hope my comments are considered constructive. Congratulations to the authors for this nice piece of work!

With kind regards.

MAJOR ISSUES

Reference to and description of the climate scenarios: On page 4 line 25 the climate scenarios are introduced as the ensemble mean of 20 GCM-RCM chains, and the term "ensemble mean" is later on frequently used. I believe this is not correct. To my knowledge the employed gridded scenarios provide three estimates for each season, each variable, each emission scenario and each grid cell: A median estimate, a lower estimate and an upper estimate. For most analyses in the present work the median estimate is used. This however is not the same as the ensemble mean signal as it originates from a probabilistic procedure that implicitly weights the underlying climate model chains. The ensemble mean field is basically only used for deriving the spatial anomalies to the regional estimates (see Zubler et al. 2014). Hence, the authors need to speak of the "median estimate" (and of the "upper" and "lower estimate" in Section 3.7). This concerns the entire manuscript.

Analysis of inter-annual variability (concerns several parts of the manuscript): In my opinion, the focus on uncertainty due to interannual variability in many analysis is not

justified. This concerns, for instance, the analysis of the d-value in Section 3.3 or the variability ranges in Figure 3 or the entire Figure S2. As the authors correctly state, the employed delta change scenarios do NOT account for changes in interannual variabililty, and the variability of the input series of temperature and precipitation will always reflect the variability of the control period. Hence, it is critical to explicitly analyse the range of signals obtained by comparing one future year to the mean state of the 13-year reference period as changes in interannual variability between the control and future period are completely neglected. My suggestion would be to rather include an assessment of climate model uncertainty (by considering always the lower, median and upper estimates of the climate scenarios).

MINOR ISSUES

Reference period: The reference period of the presented work is 1999-2012, while the reference period for the CH2011 delta change scenarios is 1980-2009, hence there's on overlap of 11 years only. This inconsistency should at least be mentioned (if not accounted for explicitly by scaling the CH2011 deltas according to difference of the mean observed climate for 1980-2009 versus 1999-2012).

Figure quality: The quality of many figures (e.g., Figs. 4, 5, 6, 7, 8, S1, S2, S4, S5, S6) is very poor (both in the PDF and when printed) and should be improved.

Aare vs Grisons: The manuscript frequently switches between showing results for the Aare region only, for Grisons only or for both. This is rather confusing, and there does not seem to be a clear motivation for this. I'd suggest to harmonize the presentation in this aspect or to better motivate the choice of one of the regions for a specific analysis.

page 2, line 14: The model employed by Marke et al. is named AMUNDSEN (nor AMSUNDSON).

page 3, line 4: "The precipitation in its northern part".

page 3, line 22: "(atmospheric)": the 3D aspect probably not only concerns the atmospheric part but also the (sub-)surface part of SNOWPACK (vertical layers in the soil and the snowpack).

page 3, line 26: The reference Bossard et al. is missing in the list of references.

page 3, lines 29-31: Unclear.

page 4, lines 11-12: The gridded climate scenarios employed here are formerly not part of the CH2011 scenarios but are an extension to them. Also they are not the only scenario product provided by the CH2011 initiative (as suggested by the second sentence). Please reformulate.

page 4, line 27: "considered for some analyses" instead of "calculated for some analysis".

page 5, line 17: "for the months January and March for the Aare region".

page 5, lines 17-18: This difference between Aare and Grisons are actually not shown, which should be indicated by adding "(not shown)" are something similar in order to avoid confusion.

page 6, lines 21-22: Can undercatch really be a reason. According to Section 2.2 undercatch of the gauges has been corrected for. It's probably more the uncertainty of such an undercatch correction that is important here.

page 6, lines 26-27: It is not clear to which metric these ranges refer. Is it the mean bias of mean winter snow depth?

page 8, line 32: "winter months".

page 9, line 9: "These results" and "who investigated".

page 9, lines 9-10: I'd doubt that it is really the temperature change anomaly that is responsible for the sensitivity of this elevation zone in terms of shortening of the snow season. It is probably the fact that this elevation range is closest to the 0°C limit and

a future temperature will hence be more effective here in terms of snow day change. At upper reaches, many parts of the winter will still remain below the freezing level. At lower reaches the snow season is anyway too short to produce important reductions in the period length of continuous snow cover.

page 9, line 12: "4.5 months".

page 9, line 16: "who demonstrate".

page 9, line 19: "who used".

page 10, line 5: "section 0" -> please correct.

page 10, line 6: Where is this inter-model variability shown? This is not clear.

page 11, line 21: "from the ensemble".

page 12, line 27: "between the individual models".

page 12, line 30: "of these projected changes".

page 13, line 31: "Rhine".

caption of Figure 3: "chapter 3.3" instead of "chapter 0".

Figure 6: In line with above comment on the analysis of interannual variability, I'd suggest to show the range of the three model uncertainty estimates for A1B. For the reference, it is OK to show the individual years.

Table S1: This table could be shortened to provide only the mean value for each site. There does not seem to be any strong trend in the RMSE scores, and the temporal evolution is anyway not discussed.

Caption of Figure S4: Please indicate that this is the Figure for the Aare Region (it is not mentioned in the caption).

Figure S5: I'd suggest to include the snow day threshold directly in the 4 panels. This

would strongly facilitate the interpretation of the figure.

---

## Referee Comment (RC2) · Anonymous Referee #2 · 4 Dec 2016

Recommendation: minor revisions

GENERAL COMMENTS

This paper represents a detailed study of possible future snow pack changes in two mountainous regions in Switzerland. The authors use a very high-resolution (200 m) surface process model (Alpine3D) specifically designed for simulation of snow conditions in complicated mountain topography. They (i) first force this model for the 1999-2012 baseline with an AWS-observation-based analysis of hourly surface weather conditions and (ii) then modify these input data using a height-sensitive Bayesian kriging analysis of RCM-simulated seasonal mean temperature and precipitation changes. By using climate changes derived from RCM simulations for three different emission scenarios and lower and upper estimates of change derived from the variation of the RCM

results, they assess the sensitive of their findings to the forcing scenario and climate modelling uncertainty.

The paper fits very well in the scope of the journal. Although there are several earlier studies on future changes in snow conditions in the European Alps and some of them use a similar methodology, this paper adds to the field (i) by providing a more comprehensive uncertainty assessment of the future changes and their sensitivity to the emission scenario, and (ii) by including a range of impact-oriented statistics, tailored to inform (e.g.) the winter tourism industry.

The paper is clearly structured and written in generally good English. However, some parts of the methodology are not very clearly described. Furthermore, the analysis of the changes in interannual variability might not be very informative because the delta change method only takes into account changes in long-term seasonal mean temperature and precipitation. Some of the figures also need improvement, particularly in the supplementary material. On the whole, however, the paper only appears to require relatively small revisions.

COMMENTS ON SCIENCE

1. Beginning of section 2.4. The method in which the climate change scenarios were obtained should be described in some more detail. From the description that is available now, the casual reader gets the impression that the RCM-simulated temperature and precipitation changes were used nearly as such. Howerer, Zubler et al. (2014) reveals that they were based on a rather complicated Bayesian methodology. In particular, the "upper and lower bounds of this dataset" are not the minima and maxima of the 20 RCM simulations, but are based on the Bayesian model of Buser et al. (2009). An important and debatable feature of this Bayesian model is that it in some cases contracts the uncertainty range from that derived directly from the variation of the original RCM simulations (see Figs. 1 and 2 in the supplementary material of Zubler et al. (2014)). Therefore, the uncertainty ranges derived in this paper should also be seen
as indicative only.

2. The simple delta change method with constant absolute changes in temperature and constant relative changes in precipitation neglects changes in climate variability on both sub-seasonal and inter-annual time scales. However, it is a common feature in climate model simulations that temperature variability in midlatitudes decreases in winter but increases in summer (e.g. Holmes et al. 2016). Such changes in climate variability almost certainly affect the change in interannual variability of snow conditions (Section 3.3). They might also have some effects on the average change in snow conditions, because the phase of precipitation and snowmelt both depend nonlinearly on temperature.

3. P12L23-24. Is this because the amount of snowfall is more sensitive to temperature when the precipitation is larger?

4. P14L2-3. Is this just common knowledge or can you support the statement with a reference?

COMMENTS ON PRESENTATION

1. P2L28-30. Why are most of the results only shown for the Aare region? This should be mentioned and motivated in the text.

2. P3L1: eastern Atlantic? (western Atlantic = east coast of North America)

3. P4L19-22. Instead of describing the socioeconomic pathways, please indicate the end-of-century $CO_2$ concentrations which give a much more tangible idea of the magnitude of climate forcing.

4. P6L7-8. "the uncertainty of model set up" should rather be "model fidelity".

5. P6L15. observed or simulated snow depth above 0.01 m?

6. P6L26 and 29. Replace "uncertainty" with (e.g.) "error" or "discrepancy"

7. P7L21-23. Would it not be simpler to directly compare the interannual variability (characterized e.g. by the coefficient of variation of snow depth or by the relative difference between the maximum and the minimum) in the baseline and future climates?

8. P8L9. What is the height range covered by the 500 m elevation zone?

9. P8L14. "The decrease of the affected years with elevation ... is caused by the lower inter-annual variability" does not make sense. The results show that the overlap between the baseline and future distributions is larger at higher elevations, most likely because the colder baseline climate make snow conditions at higher elevations less sensitive to warming.

10. P8L30. Always snow-covered in winter?

11. P8L32. winter months

12. P9L1. the fact that

13. P9L4. How do you define the beginning and end of the snow season if the first continuous snow melts away or individual days with continuous snow occur after the main snow season?

14. P9L6-7. As Figure 7 shows, the time shifts depend on elevation. What elevation do you refer to in this text?

15. P9L12 and 13. "months" in plural.

16. PL20-22. This complication could have been avoided by using the baseline period glacier mask for all periods.

17. P10L6. Note that the ...

18. P10L10-11. Please specify the emission scenario to which this result applies.

19. P11L8. Note that this is ...

20. P11L14. snow cover or snow water equivalent?

21. P13L8-9. expected snow volume reduction ... in which season?

22. P13L32. "summer months" in plural.

23. P14L9. "which might have worse consequences than the same amount of snow today"? While the preparedness will be most likely reduced, it is not obvious why the future preparedness for future's snow extremes should be worse than today's preparedness for today's more severe snow extremes.

24. Table 2. Please mention the emission scenario in the caption.

25. Figure 2. It would be simple to include the results for the Grisons region by adding a parallel set of bars with a different colour.

26. Figure 3. The headings "Aare" and "Grisons" could be inserted above the figure panels for easier reference.

27. Figure 4. How wide are the elevation zones in terms of their height ranges?

28. Figure 4. The dashed lines are too faint.

29. Figure 6. Please specify the emission scenario in the caption and increase the size of the labels.

30. Figure 8. Please increase the size of the figure panels and the size of their labels, and reduce the unnecessary empty space between them. It would also be helpful to describe the interpretation of the box plots in the caption.

31. Caption of Table S1. The values suggest that this is not a "relative error" (which would be expected to be large at stations with little snow" but rather the absolute root mean square error. In addition, please spell out the station acronyms.

32. Figure S1. Please use colours and increase the size of the headings and the labels. Also, as this figure is in the supplementary material, there is no need to squeeze the three panels on the same row.

33. Figure S2. This figure is difficult to understand. If the purpose is to illustrate the change in interannual variability, scatter plots with the yearly values of the baseline snow depth on the X axis and the corresponding future yearly values of snow depth on the Y axis would be more informative. Besides, the figure is too small.

34. Figure S4. Better in colour and with larger X and Y labels

35. Figure S6. The lines are far too faint and the labels (as well as the figure panels) far too small. Use colours for a better separation of the lines.

REFERENCES

Buser CM, Künsch HR, Lüthi D, Wild M, Schär C (2009) Bayesian multi-model projections of climate: bias assumptions and interannual variability. Clim Dyn 33:849–868.

Holmes, CR., Woollings, T., Hawkins, E, de Vries, H. (2016) Robust future changes in temperature variability under greenhouse gas forcing and the relationship with thermal advection. Journal of Climate: 29, 2221–2236

Zubler, E.M., Fischer, A.M., Liniger, M.A. et al. (2014) Localized climate change scenarios of mean temperature and precipitation over Switzerland. Climatic Change 125: 237-252

---

## Author Comment (AC1) · 23 Jan 2017

C. He
cenlinhe@atmos.ucla.edu

The authors assessed the future projection of snow depth in the Alps by accounting for future temperature and precipitation change under different emission scenarios. The results are interesting and can advance our understanding in the impact of climate change on mountain snow. Here, I have a short comment. Several recent studies (e.g., Painter et al., 2013; Liou et al., 2014; Lee et al., 2016) found that deposition of light-absorbing aerosols (mainly black carbon and dust) substantially decreases snow albedo, which further reduces snow depth and cover. However, this factor has not been considered by the authors in the future projection, which could play an important role. It would be helpful if the authors could include some discussions on these recent findings and the uncertainty due to this aerosol-snow effect in the projection of snow depth.

Response:
*In contrast to other mountain regions, the effect of the deposition of light-absorbing aerosols on the seasonal snow cover in the Alps is currently small due to the frequent snow fall events and the relatively small amount of impurities. However, we agree that this effect may be more important in future due to increasing de-glaciated area. We therefore added the following sentence to uncertainty chapter:*
*"Moreover, above this elevation glacier melt may decrease the spring snow albedo by dust from exposed moraine rubble and glacial till (Oerlemans et al., 2009)."*

---

## Author Comment (AC2) · 23 Jan 2017

The manuscript by Marty et al. is concerned with the assessment of future snow cover changes over two mountainous regions of Switzerland. For this purpose a gridded version of the CH2011 Swiss Climate Scenarios is used to force a distributed model of the surface snowpack. The analysis includes an assessment of projection uncertainties arising from the assumption of different emission scenarios and from climate model uncertainty. In agreement with previous works the study finds an important decrease of future snow cover that considerably depends on elevation and, for the late 21$^{st}$ century, on the choice of emission scenario. In a general sense, the topic of the work fits very well into the journal's scope and adds a further piece of information to 21st Century climate change impacts in the European Alps. Qualitatively and quantitatively previous works are confirmed employing a new methodology that relies on gridded climate change information and the application of a spatially distributed snow pack model. As such, I consider the work as being relevant for the scientific community. For most parts, the methods are described appropriately, and the conclusions are well-based on the results obtained. There are only minor language issues. The paper however suffers from a number of inaccuracies in the description of the underlying datasets, from a partly questionable analysis of interannual snow cover variability and from a partly poor figure quality. Please see the listing below for further details. These issues should be improved before publication of the paper. For this purpose, only few new analyses are required and the basic structure of the paper does not have to be changed. I'd therefore suggest to return the manuscript to the authors for minor revisions. I hope my comments are considered constructive. Congratulations to the authors for this nice piece of work!
With kind regards.

MAJOR ISSUES
Reference to and description of the climate scenarios: On page 4 line 25 the climate scenarios are introduced as the ensemble mean of 20 GCM-RCM chains, and the term "ensemble mean" is later on frequently used. I believe this is not correct. To my knowledge the employed gridded scenarios provide three estimates for each season, each variable, each emission scenario and each grid cell: A median estimate, a lower estimate and an upper estimate. For most analyses in the present work the median estimate is used. This however is not the same as the ensemble mean signal as it originates from a probabilistic procedure that implicitly weights the underlying climate model chains. The ensemble mean field is basically only used for deriving the spatial anomalies to the regional estimates (see Zubler et al. 2014). Hence, the authors need to speak of the "median estimate" (and of the "upper" and "lower estimate" in Section 3.7). This concerns the entire manuscript.

Response:
*We fully agree and replaced "ensemble mean" with "median estimate" throughout the manuscript.*

Analysis of inter-annual variability (concerns several parts of the manuscript): In my opinion, the focus on uncertainty due to interannual variability in many analysis is not justified. This concerns, for instance, the analysis of the d-value in Section 3.3 or the variability ranges in Figure 3 or the entire Figure S2. As the authors correctly state, the employed delta change scenarios do NOT

account for changes in interannual variabililty, and the variability of the input series of temperature and precipitation will always reflect the variability of the control period. Hence, it is critical to explicitly analyse the range of signals obtained by comparing one future year to the mean state of the 13-year reference period as changes in interannual variability between the control and future period are completely neglected. My suggestion would be to rather include an assessment of climate model uncertainty (by considering always the lower, median and upper estimates of the climate scenarios).

Response:
*We agree and therefore changed the following points:*

- *We rewrote the corresponding paragraph and added a few sentences: "The Delta change method applies changes in temperature and precipitation, which depend only on time period and emission scenario but are otherwise constant. Therefore changes in future climate variability, which may be present in the original RCM model predictions, are neglected. According to climate model projections there are no clear signs how future temperature and precipitation variability will evolve in winter in midlatitudes (Deser et al., 2012), although a recent study demonstrates a slight decrease of winter temperature variability (Holmes et al., 2016). The analyzed inter-annual variability in this study is therefore first of all determined by the inter-annual variability of the underlying temperature and precipitation conditions in the reference period. For the future scenario periods the shown inter-annual snow variability is additionally influenced by the non-linear dependence of snow on temperature, which changes the variability dependent on the size of the $\Delta T$ values."*

- *The variability information in Figure 3 was replaced as suggested with lower and upper estimates at least for one region and on emission scenario. The figure caption was changed accordingly: "Decrease of annual mean snow depth (%) relative to the reference period (1999–2012) for the Aare region and the Grisons region for the three different emission scenarios and time periods based on the median estimate change of temperature and precipitation (bars). The lowest and highest estimates (Table 3) are shown for the Aare regions and A2 scenario only (dots)."*

- *The figure caption of Figure S2 was changed to: "Distribution of the annual relative decreases of the snow depth for A2, A1B and RCP3PD and the three different future time periods (2020-49, 2045-74, 2070-99) for Aare (left) and Grisons (right) based on the inter-annual variability of the reference period."*

- *An assessment of the climate model uncertainty has now been included in Figure 6 by showing not only the median, but also the lower and upper estimates of the climate scenarios. The figure caption now reads: "Total volume of snow (Jan-Mar) in the Aare region for the today (solid line) and the end of the century (dotted line). The shaded area for the reference period indicates half of the standard deviation (for readability) of the inter-annual variability. The shaded area of the 2085 scenario period indicates the range between the lowest and highest estimate based on the A2 emission scenario (Table 3)."*

*We kept Figure S2 as a supplement since believe it provides a valuable information how the future inter-annual variability can evolve due to non-linear dependence of snow on temperature.*

MINOR ISSUES
Reference period: The reference period of the presented work is 1999-2012, while the reference period for the CH2011 delta change scenarios is 1980-2009, hence there's on overlap of 11 years only. This inconsistency should at least be mentioned (if not accounted for explicitly by scaling the CH2011 deltas according to difference of the mean observed climate for 1980-2009 versus 1999-2012).

Response:
*We agree and emphasized this fact by adding the following sentence in the Data and Methods chapter:*
*"Please note, that the reference period of these delta values (1980-2009) has an overlap of 11 years only with the reference period of the meteorological input data (1999-2012). However, a comparison of the winter temperatures for example revealed a difference of only 0.06 K between two reference periods."*

Figure quality: The quality of many figures (e.g., Figs. 4, 5, 6, 7, 8, S1, S2, S4, S5, S6) is very poor (both in the PDF and when printed) and should be improved.

Response:
*We apologize for the bad quality and included improved figures in the revised version.*

Aare vs Grisons: The manuscript frequently switches between showing results for the Aare region only, for Grisons only or for both. This is rather confusing, and there does not seem to be a clear motivation for this. I'd suggest to harmonize the presentation in this aspect or to better motivate the choice of one of the regions for a specific analysis.

Response:
*We forgot to mention this in the original manuscript. We therefore included the following sentence at the beginning of the Results chapter: "We often show results for both Alpine regions, but sometimes we focus on the Aare region only since the results are quite similar and its area below 500 m is larger and more homogeneous than the corresponding elevation zone in the Grisons region." Moreover, the results for the Grisons region were also added to Figure 2.*

page 2, line 14: The model employed by Marke et al. is named AMUNDSEN (nor AMSUNDSON).

*Corrected*

page 3, line 4: "The precipitation in its northern part".

*Corrected*

page 3, line 22: "(atmospheric)": the 3D aspect probably not only concerns the atmospheric part but also the (sub-)surface part of SNOWPACK (vertical layers in the soil and the snowpack).

*Yes, there is an optional module in ALPINE3D, which also considers the soil. However, this module was turned off in our simulations. In order to emphasize this fact we rephrased the corresponding sentence to: "It consists of a snow cover and runoff module SNOWPACK and optional modules like vegetation, soil and snow transport (Lehning et al., 2006)."*

page 3, line 26: The reference Bossard et al. is missing in the list of references.

*Corrected*

page 3, lines 29-31: Unclear.

*We agree and rephrased the sentence to: "Since this study focusses on snow on ground but not snow on glaciers, the few pixels with glacier surfaces were removed in the post-processing in order to reduce the uncertainty of our results."*

page 4, lines 11-12:
The gridded climate scenarios employed here are formerly not part of the CH2011 scenarios but are an extension to them. Also they are not the only scenario product provided by the CH2011 initiative (as suggested by the second sentence). Please reformulate.

*We agree and reformulated the sentence to: "Projections of future climate are provided as an extension of the CH2011 climate change initiative. This initiative provides among others daily change values of temperature and precipitation for Switzerland on a 2 km grid (Zubler et al., 2014),…"*

page 4, line 27: "considered for some analyses" instead of "calculated for some analysis".

*Corrected*

page 5, line 17: "for the months January and March for the Aare region".

*We actually show the mean values for period between January and March. We therefore reformulated the sentence to: "… are shown for the mean values of the January to March period."*

page 5, lines 17-18: This difference between Aare and Grisons are actually not shown, which should be indicated by adding "(not shown)" are something similar in order to avoid confusion.

*We agree and changed the sentence accordingly: "Slightly higher temperature changes in Grisons than in the Aare region are projected, especially for the end of the century (not shown)".*

page 6, lines 21-22: Can undercatch really be a reason. According to Section 2.2 undercatch of the gauges has been corrected for. It's probably more the uncertainty of such an undercatch correction that is important here.

*We agree and changed the sentence to: "High RMSE values at high-alpine sites are also explained by the fact that the measured precipitation is often heavily affected by the uncertainty of the under-catch correction and…"*

page 6, lines 26-27: It is not clear to which metric these ranges refer. Is it the mean bias of mean winter snow depth?

*We agree and clarified the sentence to: "…the uncertainty in simulating the mean winter snow depth in the reference period at the point scale (between -15 and 26 % for the different stations) increases…"*

page 8, line 32: "winter months".

*corrected*

page 9, line 9: "These results" and "who investigated".

*corrected*

page 9, lines 9-10: I'd doubt that it is really the temperature change anomaly that is responsible for the sensitivity of this elevation zone in terms of shortening of the snow season. It is probably the fact that this elevation range is closest to the 0 C limit and a future temperature will hence be more effective here in terms of snow day change. At higher reaches, many parts of the winter will still remain below the freezing level. At lower reaches the snow season is anyway too short to produce important reductions in the period length of continuous snow cover.

We agree and changed the sentence to: "*This is probably caused by the fact that this elevation zone is closest to the 0°C limit. At upper reaches, many parts of the winter will still remain below the freezing level. At lower reaches the snow season is anyway too short to produce important reductions in the period length of continuous snow cover.*"

page 9, line 12: "4.5 months".

*corrected*

page 9, line 16: "who demonstrate".

*corrected*

page 9, line 19: "who used".

*corrected*

page 10, line 5: "section 0" -> please correct.

*corrected*

page 10, line 6: Where is this inter-model variability shown? This is not clear.

*We agree, this is not shown. We therefore reformulated the sentence to: "Note that the, the inter-model variability, from which the median estimate is calculated, is much smaller than the inter-annual variability as shown in Schmucki et al. (2015b)."*

page 11, line 21: "from the ensemble".

*corrected*

page 12, line 27: "between the individual models".

*corrected*

page 12, line 30: "of these projected changes".

*corrected*

page 13, line 31: "Rhine".

*corrected*

caption of Figure 3: "chapter 3.3" instead of "chapter 0".

*The figure has been changed. A correction is therefore not anymore necessary.*

Figure 6: In line with above comment on the analysis of interannual variability, I'd suggest to show the range of the three model uncertainty estimates for A1B. For the reference, it is OK to show the individual years.

*We agree and implemented the suggested changes.*

Table S1: This table could be shortened to provide only the mean value for each site. There does not seem to be any strong trend in the RMSE scores, and the temporal evolution is anyway not discussed.

*This could be done, but we prefer the current version because the values for the individual years indeed provide the information that the error is not dependent on snow abundant or snow scarce years.*

Caption of Figure S4: Please indicate that this is the Figure for the Aare Region (it is not mentioned in the caption).

*corrected*

Figure S5: I'd suggest to include the snow day threshold directly in the 4 panels. This would strongly facilitate the interpretation of the figure.

*implemented*

---

## Author Comment (AC3) · 23 Jan 2017

Recommendation: minor revisions

GENERAL COMMENTS
This paper represents a detailed study of possible future snow pack changes in two mountainous regions in Switzerland. The authors use a very high-resolution (200 m) surface process model (Alpine3D) specifically designed for simulation of snow conditions in complicated mountain topography. They (i) first force this model for the 1999-2012 baseline with an AWS-observation-based analysis of hourly surface weather conditions and (ii) then modify these input data using a height-sensitive Bayesian kriging analysis of RCM-simulated seasonal mean temperature and precipitation changes. By using climate changes derived from RCM simulations for three different emission scenarios and lower and upper estimates of change derived from the variation of the RCM results, they assess the sensitive of their findings to the forcing scenario and climate modelling uncertainty.

The paper fits very well in the scope of the journal. Although there are several earlier studies on future changes in snow conditions in the European Alps and some of them use a similar methodology, this paper adds to the field (i) by providing a more comprehensive uncertainty assessment of the future changes and their sensitivity to the emission scenario, and (ii) by including a range of impact-oriented statistics, tailored to inform (e.g.) the winter tourism industry.

The paper is clearly structured and written in generally good English. However, some parts of the methodology are not very clearly described. Furthermore, the analysis of the changes in interannual variability might not be very informative because the delta change method only takes into account changes in long-term seasonal mean temperature and precipitation. Some of the figures also need improvement, particularly in the supplementary material. On the whole, however, the paper only appears to require relatively small revisions.

COMMENTS ON SCIENCE
1.  Beginning of section 2.4. The method in which the climate change scenarios were obtained should be described in some more detail. From the description that is available now, the casual reader gets the impression that the RCM-simulated temperature and precipitation changes were used nearly as such. However, Zubler et al. (2014) reveals that they were based on a rather complicated Bayesian methodology. In particular, the "upper and lower bounds of this dataset" are not the minima and maxima of the 20 RCM simulations, but are based on the Bayesian model of Buser et al. (2009). An important and debatable feature of this Bayesian model is that it in some cases contracts the uncertainty range from that derived directly from the variation of the original RCM simulations (see Figs. 1 and 2 in the supplementary material of Zubler et al. (2014)). Therefore, the uncertainty ranges derived in this paper should also be seen as indicative only.

*We thank the reviewer for this valuable comment. We changed to corresponding section accordingly: "The focus of this work is related to the (ensemble) mean median estimate of these 20 different combinations, which were derived by Bayesian methodology. The upper and lower bounds estimates (extremes) of this dataset, which contains the 97.5 %, respectively 2.5 % quantile of the 20 member ensembles are also considered for some analyses calculated for some analysis in order to get information about the range of the uncertainties of the temperature*

*and precipitation change. Hereof, it is important to know that this Bayesian methodology contracts in some cases the uncertainty range directly derived from the variation of the original RCM simulations. Therefore, the uncertainty range in this paper should also be seen as indicative only. A simple delta change approach was used to compile meteorological time series of future Alpine climate. This means that the time series of the reference period were modified with the provided gridded daily change values of the air temperature ($\Delta T$) and precipitation ($\Delta P$). More information about the calculation of these delta values and about the downscaling and can be found in Zubler et al. (2014).”*

2. The simple delta change method with constant absolute changes in temperature and constant relative changes in precipitation neglects changes in climate variability on both sub-seasonal and inter-annual time scales. However, it is a common feature in climate model simulations that temperature variability in midlatitudes decreases in winter but increases in summer (e.g. Holmes et al. 2016). Such changes in climate variability almost certainly affect the change in interannual variability of snow conditions (Section 3.3). They might also have some effects on the average change in snow conditions, because the phase of precipitation and snowmelt both depend nonlinearly on temperature.

*We thank the reviewer for this comment and the corresponding reference. However, we are not convinced that the decrease of winter temperature variability in midlatitudes is a common feature in climate model simulations. Moreover, the evolution of the winter precipitation variability seems to be even less clear. We tried to explain that in few extra sentences and additionally emphasized the we cannot analyze the inter-annual variability as simulated by the RCM's with the applied method: “Since the applied Delta change method with scenario and time period dependent constant changes in temperature and precipitation neglects changes in future climate variability, the shown inter-annual snow variability cannot mirror the simulated inter-annual variability of the RCMs. According to climate model projections there are no clear signs how future temperature and precipitation variability will evolve in winter in midlatitudes (Deser et al., 2012), although a recent study demonstrates a slight decrease of winter temperature variability (Holmes et al., 2016). The analyzed inter-annual variability in this study is therefore first of all determined by the inter-annual variability of the underlying temperature and precipitation conditions in the reference period. For the future scenario periods the shown inter-annual snow variability is additionally influenced by the non-linear dependence of snow on temperature, which changes the variability dependent on the size of the ΔT values.”*

3. P12L23-24. Is this because the amount of snowfall is more sensitive to temperature when the precipitation is larger?

*We don't know, but that could be a possible explanation, since larger precipitation events often happen at relatively warm temperatures.*

4. P14L2-3. Is this just common knowledge or can you support the statement with a reference?

*At least for "snow" people it is common knowledge, nevertheless we added the following references for those less familiar with this issue:*
- *Norrman, J., Eriksson, M., and Lindqvist, S.: Relationships between road slipperiness, traffic accident risk and winter road maintenance activity, Climate Research, 15, 185-193, 2000.*

- *Hess, M., Saska, M., and Schilling, K.: Application of coordinated multi-vehicle formations for snow shoveling on airports, Intelligent Service Robotics, 2, 205, 10.1007/s11370-009-0048-5, 2009.*
- *Schmidlin, T. W.: Impacts of Severe Winter Weather during December 1989 in the Lake Erie Snowbelt, Journal of Climate, 6, 759-767, 10.1175/1520-0442(1993)006<0759:ioswwd>2.0.co;2, 1993.*

COMMENTS ON PRESENTATION

1. P2L28-30. Why are most of the results only shown for the Aare region? This should be mentioned and motivated in the text.

*We forgot to mention this in the original manuscript. We therefore included the following sentence at the beginning of the Results chapter: "We often show results for both Alpine regions, but sometimes we focus on the Aare region only since the results are quite similar and its area below 500 m is larger and more homogeneous than the corresponding elevation zone in the Grisons region." Moreover, the results for the Grisons region were also added to Figure 2.*

2. P3L1: eastern Atlantic? (western Atlantic = east coast of North America)

*We agree and changed the sentence to: "The precipitation amount in this region is mainly controlled by large scale weather patterns coming from the northern Atlantic."*

3. P4L19-22. Instead of describing the socioeconomic pathways, please indicate the end-of-century $CO_2$ concentrations which give a much more tangible idea of the magnitude of climate forcing.

*We added the end of century $CO_2$ concentrations and changed the sentences to: "The A1B scenario is characterized by a rapid economic growth with a mixture of fossil and non-fossil energy sources.The maximum population will peak around 2050 and the CO2 concentration is roughly 720 ppm at end of the century. In the A2 scenario a continuously increasing population and a low economic rate of growth is assumed and the CO2 concentration reaches roughly 860 ppm (Nakicenovic and Swart, 2000)."*

4. P6L7-8. "the uncertainty of model set up" should rather be "model fidelity".

*corrected*

5. P6L15. observed or simulated snow depth above 0.01 m?

*The observed snow depth. The sentence reads now: "The RMSE was calculated for each of the 13 years of the reference period for the observed snow depth above 0.01 m (Table S1)."*

6. P6L26 and 29. Replace "uncertainty" with (e.g.) "error" or "discrepancy"

*corrected*

7.  P7L21-23.  Would it not be simpler to directly compare the interannual variability (characterized e.g. by the coefficient of variation of snow depth or by the relative difference between the maximum and the minimum) in the baseline and future climates?

*Yes, that would be another possibility, but this approach would not be able to provide the valuable information how the future inter-annual variability changes due to non-linear dependence of snow on temperature.*

8.  P8L9. What is the height range covered by the 500 m elevation zone?

*The height range is a 100 m band, i.e. between 450 and 550 m for the 500 m elevation zone. We indicated this by changing the sentence to: "At the 500 m (450-550m) elevation zone..."*

9.  P8L14. "The decrease of the affected years with elevation ...   is caused by the lower inter-annual variability" does not make sense. The results show that the overlap between the baseline and future distributions is larger at higher elevations, most likely because the colder baseline climate make snow conditions at higher elevations less sensitive to warming.

*We agree and changed the sentence to: "At these higher elevations more winters remain with maximum snow depths higher than the current minimal snow depth. This is caused by the fact that the colder baseline climate makes snow conditions at higher elevations less sensitive to warming."*

10.  P8L30. Always snow-covered in winter?

*We agree and changed the sentence to: "The absolute decrease is largest between 1500 and 2500 m asl, since this elevation band is nowadays always snow covered during the winter months and heavily affected by warmer temperatures."*

11.  P8L32. winter months

*corrected*

12.  P9L1. the fact that

*corrected*

13.  P9L4.  How do you define the beginning and end of the snow season if the first continuous snow melts away or individual days with continuous snow occur after the main snow season?

*The beginning and end of the snow season was defined as the longest snow covered period in each season. We there changed the sentence to: "The date of the first continuous snow (snow depth at least 1 cm) and the end of the snow season was calculated based on the longest snow covered period for each of the 13 years for all time periods."*

14.  P9L6-7.  As Figure 7 shows, the time shifts depend on elevation.  What elevation do you refer to in this text?

*The former text refered to the elevation zone between 1000 and 2000 m. We realized and that the differences within this zone are large. Therefore we refer now to 1500 m and changed the text accordingly: "At 1500 m, for example, the snow season starts on average about 2 (2035) to 5 (2085) weeks later and ends 2 (2035) to 12 (2085) weeks earlier."*

15.  P9L12 and 13. "months" in plural.

*corrected*

16.  P9L20-22. This complication could have been avoided by using the baseline period glacier mask for all periods.

*Yes, you're right, that would have been another possibility.*

17.  P10L6. Note that the ...

*corrected*

18.  P10L10-11. Please specify the emission scenario to which this result applies.

*We added this information, which changed the sentence to: "During the middle of the century and the A2 emission scenario, the same probability can be found at 850 m asl."*

19.  P11L8. Note that this is ...

corrected

20.  P11L14. snow cover or snow water equivalent?

*corrected to snow water equivalent*

21.  P13L8-9. expected snow volume reduction ... in which season?

*We added the information and changed the sentence accordingly: "Both regions show a similar clear reduction in the future snow volume (Jan-Mar)…"*

22.  P13L32. "summer months" in plural.

corrected

23. P14L9. "which might have worse consequences than the same amount of snow today"? While the preparedness will be most likely reduced, it is not obvious why the future preparedness for future's snow extremes should be worse than today's prepared- ness for today's more severe snow extremes.

*We see a clear difference in this respect. Today, we can also handle extreme snow amounts, , because we have the tools and some experience from "normal" snow events. In future, we will be much less prepared for snow abundant winters, because we don't have the experience and infrastructure anymore, since we are used to winters without snow. An analogue of such situations can be seen in the rare cases, when there is abundant snow in such places as the Mediterranean or Florida. Nevertheless, we rephrased the corresponding paragraph: "Due to fact that precipitation towards the end of the century is projected to slightly increase, it is therefore probable, that also in future (then) unusual winters may experience short periods of abundant snow, which might have worse consequences than today since society probably would be less prepared for such rare events"*

24. Table 2. Please mention the emission scenario in the caption.

*We added the information and changed the caption accordingly: "Number of days with more than 30 cm/50 cm of snow depth at 3000 m asl in the Aare region for the reference period and the three future scenario periods based on the A2 emission scenario."*

25. Figure 2. It would be simple to include the results for the Grisons region by adding a parallel set of bars with a different colour.

*We agree and added the results for the Grisons region.*

26. Figure 3. The headings "Aare" and "Grisons" could be inserted above the figure panels for easier reference.

*Is not necessary since the figure has been redesigned anyway and the two regions have been included in the legend.*

27. Figure 4. How wide are the elevation zones in terms of their height ranges?

*We added the following sentence to figure caption: "The elevation zones are 100 m wide, i.e. the 1500 m zone, contains all pixels between 1450 and 1550 m."*

28. Figure 4. The dashed lines are too faint.

*corrected*

29. Figure 6. Please specify the emission scenario in the caption and increase the size of the labels.

*The size of the labels has been increased and the caption reads now: "Total volume of snow (Jan-Mar) in the Aare region for the today (solid line) and the end of the century (dotted line) based on the A2 emission scenario."*

30.  Figure 8. Please increase the size of the figure panels and the size of their labels, and reduce the unnecessary empty space between them.  It would also be helpful to describe the interpretation of the box plots in the caption.

*Corrected and the following description of the box plots has been added to the figure caption: "The little square in the box plots represents the mean value and the whiskers show the 2.5 % and 97.5 % quantile value of the different model simulations."*

31.  Caption of Table S1.  The values suggest that this is not a "relative error" (which would be expected to be large at stations with little snow" but rather the absolute root mean square error. In addition, please spell out the station acronyms.

*Yes, you're absolutely right. We therefore changed the table caption accordingly and also replaced the stations acronyms with their names.*

32. Figure S1. Please use colours and increase the size of the headings and the labels. Also, as this figure is in the supplementary material, there is no need to squeeze the three panels on the same row.

*corrected*

33.  Figure S2.  This figure is difficult to understand. If the purpose is to illustrate the change in interannual variability,  scatter plots with the yearly values of the baseline snow depth on the X axis and the corresponding future yearly values of snow depth on the Y axis would be more informative. Besides, the figure is too small.

*We produced a scatter plot figure as suggested, but we cannot see the scientific gain of such an approach. The original figure has therefore been enlarged but we agree is still not easy to understand. However, we believe the figure provides a good approach to demonstrate how the distribution of snow abundant and snow scare winters changes dependent on the time and emission scenario due to non-linear dependence of snow on temperature.*

34. Figure S4. Better in colour and with larger X and Y labels

*corrected*

35.  Figure S6.  The lines are far too faint and the labels (as well as the figure panels) far too small. Use colours for a better separation of the lines.

*corrected*

---

## Author Response (AR2)

**tc-2016-230:  Response to editor (2017-01-25)**

*To scientific editor (Ross Brown)*

5 *We thank the editor for his comments and changed the few sentences as suggested. The manuscript with track changes can be found below.*

*Best regards*

*Christoph Marty*

===========================================================

Editor Decision: Publish subject to minor revisions (Editor review) (24 Jan 2017) by Mr Ross Brown

15 Comments to the Author:

Hi Christoph - from my reading, the revised m/s addresses all the major points raised by the reviewers. However, there are a few places in the paper where some phrases are rather difficult to understand. I've highlighted a few of the obvious ones below. I recommend you ask an anglophone colleague or technical editor to proof-read the manuscript to streamline the text

20 and improve readability.
Best regards, Ross B

*We thank the editor for his comments and changed the few sentences as suggested. The text has now been proof-read by an Anglophone collaborator, which caused a lot word exchanges*

25 *as can be seen in the manuscript with track changes below.*

*Best regards*

*Christoph Marty*

1. In Section 3.5 the following addition does not read clearly:
"This is probably caused by the fact that this elevation zone is then closest to the 0°C limit. At higher reaches, many parts of the winter will still remain below the freezing level. At lower reaches the snow season is anyway too short to produce important reductions in the period

35 length of continuous snow cover."

I suggest rewording along the lines of:

"This is probably caused by this elevation zone being closest to the 0°C limit. At higher elevations, air temperatures remain below-freezing for most of the winter period, while at lower elevations, the snow season is too short to for warming to generate large reductions in the period of continuous snow cover."

*We agree and changed the sentence to: "This is probably caused by this elevation zone being closest to the 0°C limit. At higher elevations, air temperatures remain below the freezing level for most of the winter period, while at lower elevations the snow season is too short for warming to generate large reductions in the period of continuous snow cover."*

2. The following sentence in Section 3.7 is difficult to follow:
"Moreover, above this elevation glacier melt may decrease the spring snow albedo by dust from exposed moraine rubble and glacial till..."

I think you mean that increasing exposure of rubble and till from glacier melt augments the potential for deposition of dust on the glacier surface, contributing to a lowered surface albedo and a positive feedback that is not taken into account in the present modelling study.

*We agree and changed the sentence to: "Moreover, above this elevation increasing exposure of rubble and till from glacier melt augments the potential for deposition of dust on the glacier surface, contributing to a lowered surface albedo and a positive feedback (Oerlemans et al., 2009), which is not taken into account in our study."*

3. The final sentence in the paper is not well-constructed and does not really follow from the preceding discussion. You also don't look at extreme snowfall events in this study so this final closing statement is not a strong way to finish. I suggest you remove it.

*We removed the sentence.*

[revised manuscript text omitted]

10   in the same region (mainly dependent on elevation) and must be considered. Finally, the $\Delta T$ is added to the air temperature time series  and the $\Delta P$ is multiplied by the precipitation time series  for each year of the reference period  using a simple delta approach.

$$T_{scen} = T_{ref} + \Delta T \qquad (1)$$

15  $$P_{scen} = P_{ref} \bullet (1+\Delta P) \qquad (2),$$

where $\Delta P$ is given in [%]. A time series of $\Delta T$ and $\Delta P$ shows the seasonal variations of the climate signals. The highest median estimate of $\Delta T$ was clearly found for summer. The lowest $\Delta T$ is predicted for spring, with only slightly higher changes in winter and autumn. Depending on the emission scenarios, the seasonal range of the median

20  estimate of $\Delta T$ varies between 0.4 and 1.0°C for the end of the century . In contrast to temperature, the change in the seasonal precipitation can only be predicted with high uncertainties. The projection range of the 20 different climate models covers decreasing and increasing precipitation for almost all seasons and scenario periods. The  median estimate of precipitation changes significantly only in summer time, with a decrease of up to 30% towards the end of the century. In spring the precipitation is predicted to increase by up to 10%. For winter and autumn the

25  precipitation will not change significantly. When investigating snow related questions, the climate change signals for winter and the beginning of spring are more important than changes in summer time.

In Figure 2 the median estimate deltas and their uncertainty range are shown for the means of the months January to March. Slightly higher temperature changes in Grisons than in the Aare region are projected, especially for the end of the century. In this scenario period  precipitation increases according to the A2 scenario by 4.3% in the Aare region and 7.3% in Grisons

30  . The influence of the precipitation change is negligible compared to the temperature changes, because the predicted changes in precipitation are very small in the winter half year (Schmucki et al., 2015a).

Due to the fact that the parameterized incoming longwave radiation (ILWR) is a function of  temperature, we calculated the parameterization of the ILWR for each emission scenario separately. This implies an emission scenario dependent

ILWR, which is necessary because  ILWR fluxes contribute significantly to  snow melt, especially in spring (Schlögl et al., 2016).

Changes of glacier coverage were provided by Linsbauer et al. (2013) and used in order to adapt the land cover data to future scenarios. The changes were calculated with an elevation dependent ice thickness model (M2) for the three emission
5   scenarios and the three different time periods. Future glacier free areas were assumed as pixels with rocks in the land cover data. Note that the ice thickness model is only forced by the temperature change. Changes in the precipitation as seen in Figure 2 were neglected in the model because the  uncertainties in the assessment of future precipitation are too high. The current relative amount of glacial areas in the Aare region (6.7%) is higher than in Grisons (1.7%). The future glacier covered area will be halved until 2060 and only a few pixels will still be covered with glaciers towards the end of the
10   century.

**3 Results and Discussion**

We present projected changes of snow depth and duration for two Alpine regions based on the difference between the simulated values of the reference period and 9 different climate projections (3 time periods and 3 emission scenarios). The
15   results are mainly based on the median estimate of all 20 model combinations, however in the last paragraph the uncertainty based on the 95% spread (upper and lower estimates) is also shown. We often show results for both Alpine regions, but sometimes we focus on the Aare region only since the results are quite similar and its area below 500 m elevation is larger and more homogeneous than the corresponding elevation zone in the Grisons region.

**3.1 Validation**

20   By comparing the modelled Alpine3D snow depths of the reference period with measured snow depths, the model fidelity is estimated by means of the RMSE. The nearest and all neighbouring pixels (9 in total) were considered for comparison with the station value. The pixel which showed the best agreement with the station elevation was chosen for comparison. The agreement is generally good, but  in a heterogeneous topography like the Alps such a comparison will always be limited by the fact that the observations are point measurements in a flat field and the pixel value
25   represents an average over an area which is inclined and at a different elevation. Moreover, measured snow depth in high-alpine flat fields usually is higher than the spatially averaged snow depth, e.g. from a grid cell (Grünewald and Lehning, 2015) and therefore generally not representative of a larger area. The RMSE was calculated for each of the 13 years of the reference period for the observed snow depth above 0.01 m (Table S1). Mountain stations generally show a higher RMSE due to the topographical effect described above. Figure S1 illustrates some typical cases, where the
30   simulated snow cover is either too  great or too small: At the high  elevation station Weissfluhjoch (2540 m) in the Grisons region the simulation underestimates  
[revised manuscript text omitted]
  roughly at 1500 m asl. This is probably caused by  this elevation zone being  closest to the 0°C limit. At higher elevations, air temperatures  remain below the freezing level for most of the winter period, while at

lower  elevations the snow season is too short for warming to generate large  reductions in the period  of continuous snow cover. These results confirm the findings of Kotlarski et al. (2015), who investigated the elevation dependency of the number of snow days in 5 RCMs and found a maximum reduction for the winter half year at about 1500 m asl.

The snow season at 1000 m asl currently lasts about 4 months from December until the end of March. At the end of the century almost no snow is projected at this elevation. A similar reduction of 4.5 months can be observed at 1500 m asl, where the duration of continuous snow cover is reduced to only 2 months, i.e. mid  December to mid  February. It should be noted that these numbers are based on an average winter in the corresponding time period and neglect the fact that future winters  at this elevation will often be characterized by ephemeral snow cover, which is nowadays only typical for elevations below 1000 m . This result is in good agreement with the findings of Schmucki et al. (2015b), who demonstrate that at 1500 m asl in the Swiss Alps the probability for the occurrence of a winter with a continuous snow cover is only 60% at the end of the century. Generally, the decrease in snow duration is equal to an elevation shift of 200-500 m for the first scenario period and 700-1000 m for the last scenario period for the A2 scenario. This  is in agreement with a study  by Bavay et al. (2013), who  used 3 RCMs  and found similar  values for the Swiss Alps. The slight bump at 2800 m in the curve of the first scenario period (Figure 7) results from the  lower glacier coverage in the  period 2020-2049. Originally deleted pixels (due to glacier coverage) are now snow covered pixels in this time period (see 2.3).

**3.6 Number of snow days**

The demonstrated decrease in snow depth and snow duration also affects  the number of snow days. We define a snow day as a day with a least 5 cm snow on the ground, because with  regard to winter tourism this is the minimum snow depth to generate a winter feeling,  build a snow-man, or  go sledding. The number of  snow days was  calculated for the four time periods for several towns in the two investigated regions. Table S2 shows the median number of such snow days for the A2 scenario. The results clearly show that the number of snow days  on the Swiss plateau will  reach zero in the  final scenario period. A multi-day snow cover will therefore be a rare event towards the end of the century in this elevation zone. Stations at about 1500 m will lose ca. 100 snow days, especially in the melting season. Davos (1560 m), for example, will only have 10 snow days more at the end of the century than Chur (593 m) has today and Adelboden (1350 m) will  have less snow days than Bern (542 m) has at present .

The inter-annual variability in snow days is shown in Figure 8 for three selected stations in the Aare region. The range is highest for Bern (542 m) at  present , for Grindelwald (1034 m) in the first scenario period and for Mürren (1650 m) in the last scenario period, which corresponds well with the findings described in section 3.3, where the elevation with the

highest variability increased with time. Note that the inter-model variability, from which the  median estimate is calculated, is much  lower than the inter-annual variability as shown  by Schmucki et al. (2015b).

The  probabilities of a winter with 0 snow days, less than 5, 15 or 50 snow days depending on elevation and scenario period  are shown Figure S5. As expected,  the same probability in future would be found at higher elevation. For example,  there is a 7% probability that we experience less than 5 snow days at 500 m asl today.  During the middle of the century and using the A2 emission scenario the same probability can be found at 850 m asl.

[revised manuscript text omitted]